# Pollinator sharing, copollination, and speciation by host shifting among six closely related dioecious fig species

Zhi-Hui Su [1,2✉], Ayako Sasaki[1], Junko Kusumi [3], Po-An Chou[4], Hsy-Yu Tzeng[4], Hong-Qing Li[5] & Hui Yu[6]

The obligate pollination mutualism between figs (*Ficus*, Moraceae) and pollinator wasps (Agaonidae, Hymenoptera) is a classic example of cospeciation. However, examples of phylogenetic incongruencies between figs and their pollinators suggest that pollinators may speciate by host shifting. To investigate the mechanism of speciation by host shifting, we examined the phylogenetic relationships and population genetic structures of six closely related fig species and their pollinators from southern China and Taiwan-Ryukyu islands using various molecular markers. The results revealed 1) an extraordinary case of pollinator sharing, in which five distinct fig species share a single pollinator species in southern China; 2) two types of copollination, namely, sympatric copollination by pollinator duplication or pollinator migration, and allopatric copollination by host migration and new pollinator acquisition; 3) fig species from southern China have colonized Taiwan repeatedly and one of these events has been followed by host shifting, reestablishment of host specificity, and pollinator speciation, in order. Based on our results, we propose a model for pollinator speciation by host shifting in which the reestablishment of host-specificity plays a central role in the speciation process. These findings provide important insights into understanding the mechanisms underlying pollinator speciation and host specificity in obligate pollination mutualism.

[1] JT Biohistory Research Hall, Takatsuki, Osaka 569-1125, Japan. [2] Department of Biological Sciences, Graduate School of Science, Osaka University, Osaka 560-0043, Japan. [3] Department of Environmental Changes, Faculty of Social and Cultural Studies, Kyushu University, Fukuoka 819-0395, Japan. [4] Department of Forestry, National Chung Hsing University, Taichung 402, Taiwan. [5] School of Life Sciences, East China Normal University, Shanghai 200241, China. [6] Key Laboratory of Plant Resource Conservation and Sustainable Utilization, South China Botanical Garden, The Chinese Academy of Sciences, Guangzhou 510650, China. ✉email: su.zhihui@brh.co.jp

Interactions between angiosperms and insect pollinators are believed to have contributed to the considerable species diversity that is evident in both groups[1–6]. Obligate pollination mutualism is an extreme example of plant-insect interactions, and one of the classic examples of such interactions is the association between figs (*Ficus*, Moraceae) and their wasp pollinators (Agaonidae, Hymenoptera)[7,8]. Figs depend on wasps for pollination and the wasps depend on figs to complete their life cycle by providing sites for oviposition, larval hatching, growth, development and reproduction. Indeed, all of these stages in the life history of wasps take place within the closed inflorescence (syconium) of a fig[7,9]. While it may seem that obligate pollination mutualism would limit the speciation of hosts and pollinators, *Ficus* is one of the most widespread and speciose plant genera in tropical and subtropical regions, with more than 750 described species[10–12]. It has been estimated that approximately the same number of wasp species pollinate figs, and of these, 319 species have been described to date[12]. Investigation of speciation mechanisms associated with fig-wasp mutualism may help us to understand the high levels of species diversity in both groups.

The obligate mutualism between figs and pollinator wasps has long provided an ideal framework for addressing questions related to coevolution and cospeciation between insects and plants[8,9,13], since it has been considered that this form of mutualism is characterized by high host-specificity, i.e., the "one to one" rule, where one fig species is pollinated by only one species of wasp[7,14]. However, recent studies have demonstrated numerous instances of co-pollination, i.e., where one fig species is associated with more than one pollinator species, as well as some examples of pollinator sharing, i.e., where one pollinator species is associated with more than one fig species[8,13,15–26]. The most significant implication of these findings is that numerous pollinator species and some fig species have not arisen by strict cospeciation[13]. Moreover, while molecular phylogenetic analyses broadly support the co-cladogenesis in figs and pollinator wasps at higher taxonomic levels[27,28], the phylogenies at the species level of figs and wasps do not match perfectly; indeed, the existence of such incongruencies[8,28–32] suggests that mechanisms other than strict cospeciation also play a significant role in the species diversity[13]. In response to these reports, it has been suggested that pollinator wasps may speciate by host shifting between closely related figs, or by duplication (i.e., where wasps speciate within the same host fig species, and the newly generated co-pollinators have a sister relationship in terms of phylogeny)[13]. However, little is known about the mechanisms underlying these speciation processes; for example, which ecological conditions favour host shifting or duplication, and which speciation mechanism, sympatric or allopatric, is employed during host shifting and duplication. Cook and Segar[13] proposed that the duplication mechanism of pollinator speciation comprised three main steps: first, fig and wasp populations split into geographically and genetically isolated sub-populations; second, divergence occurs between the sub-populations, but proceeds more rapidly in wasps than in figs; third, on secondary contact between the sub-populations, the figs, but not the wasps, can interbreed which results in the co-occurrence of the two wasps (co-pollinators). Under this scenario, geographic isolation appears to be the key factor underlying the duplication in pollinators. In this study, we hypothesize that host shifting acts as a trigger that leads to the sympatric speciation of pollinators through the reestablishment of host specificity. The main aim of this study was to find support for this hypothesis by examining the phylogenetic relationships between figs and their pollinators in areas where host shifting and host-specificity reestablishment are expected to occur.

Most fig species (ca. 511 species) are distributed in the Indo-Australasian region, with the centre of Malesia being a hotspot of species richness (359 species)[12]. Some of the fig species in this region have spread northward to mainland China, Taiwan, and Japan, including the Ryukyu Islands. For species in highly species-specific mutualistic relationships, long-distance island colonization may disrupt their mutualistic relationships due to the need for synchronous co-dispersal of both partners. However, in cases where only one of the partners in a mutualistic relationship colonize an island, the possibility exists that new mutualistic relationships can be established if suitable partners can be found[33,34]. It is expected that such colonization events have occurred repeatedly during the northward range expansion of fig-wasp mutualism to Taiwan and the Ryukyu Islands, which represent the northernmost boundary of the *Ficus* distribution range. In addition, molecular phylogenetic analyses have suggested that co-pollinator and pollinator sharing, which possibly arose by host shifting, have arisen in dioecious figs in Taiwan[18,24]. Consequently, the figs and pollinator wasps distributed throughout the Taiwan-Ryukyu islands appear to be well suited to elucidating how pollinator sharing, co-pollination, and host-shift speciation arise in fig-wasp mutualism.

To date, examples of pollinator sharing and host shifting have been observed in closely related fig species[22–25,35,36]. Therefore, as materials for this study, we selected six very closely related fig species from the *F. erecta* complex, which comprises ~17 species[37] that are distributed in southern China and/or Taiwan-Ryukyu islands. We clarified the population genetic structure and detailed phylogenetic relationships among the six fig species and their pollinator wasps using various molecular data, with extensive sampling from the aforementioned distribution areas (Fig. 1, Table 1, and Supplementary Data 1). The results revealed an extraordinary example of pollinator sharing in which five distinct fig species share a single pollinator species, that fig colonization occurred repeatedly from mainland China to Taiwan, and that the occurrence of co-pollinators and pollinator speciation occurred by host shifting. Our findings provide important insights into understanding the mechanisms of pollinator speciation and host specificity in obligate pollination mutualism.

## Results

**Phylogenetic relationships of figs based on MIG-seq data.** Molecular markers, such as chloroplast DNA, internal transcribed spacer (ITS) regions of the nuclear rRNA gene, *1-aminocyclopropane-1-carboxylate oxidase* (*aco1*), and *glyceraldehyde 3-phosphate dehydrogenase* (*g3pdh*) are often used in phylogenetic analyses of fig plants[16,27,28,30,36,37]. However, due to the limited phylogenetic information that can be obtained from these markers at the species level, it is difficult to clarify the phylogenetic relationships among closely related figs. Therefore, in this study, we used multiplexed ISSR genotyping by sequencing (MIG-seq) data to analyze the phylogenetic relationships among six closely related fig species, because the MIG-seq data are expected to provide finer taxonomic resolution for these fig species. The average number of raw and high quality reads obtained from 60 fig individuals belonging to eight species was 1,133,084 (ranging from 712,862 to 1,593,488) and 491,420 (ranging from 321,948 to 719,792), respectively. De novo assembly yielded 80,138 loci, of which 5284 were polymorphic. After removing the non-phylogenetically informative sites, the final data set for phylogenetic analyses produced 2198 variable nucleotide sites. The maximum likelihood (ML) and Bayesian inference (BI) trees (Fig. 2 and Supplementary Fig. 1) showed that all of the fig species formed monophyletic groups with strong bootstrap support, but with low support for *Ficus erecta*. Four populations of *F. erecta* were clustered into two groups; one

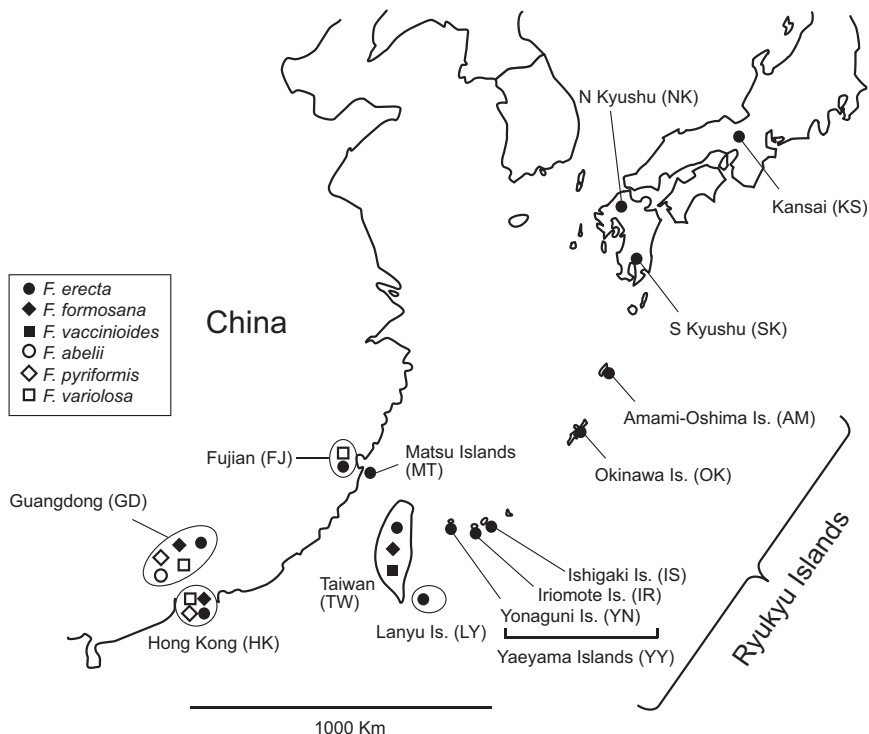

**Fig. 1 Sampling locations of figs and their pollinator wasps used in this study.** The abbreviations of population names are given in parentheses and correspond to those used in other figures and Tables.

| Table 1 Samples and molecular data used in this study. | | | | | | | | | | | | | |
|---|---|---|---|---|---|---|---|---|---|---|---|---|---|
| **Host figs** | | **Pollinator wasps** | | **Population** | **Number of individuals analyzed for figs** | | | | **Number of individuals analyzed for pollinators** | | | | |
| **Species** | **No. of individuals** | **Species** | **No. of individuals** | | **Plastid** | **ITS** | **Mig-seq** | **SSR** | **28 S** | **ITS1** | **COI–COII** | **SSR** | |
| *F. erecta* | 14 | *B. nipponica* | 17 | KS, NK, SK, AM, OK | 14 | | | | 17 | | 17 | | |
| | 39 | *B. nipponica* | 45 | IS | 14 | | 5 | 38 | 3 | 2 | 43 | 42 | |
| | 36 | *B. nipponica* | 18 | IR | 22 | | | 36 | 2 | | 16 | 14 | |
| | 35 | *B. nipponica* | 26 | YN | 35 | | | 35 | 1 | | 26 | 25 | |
| | 48 | *B. nipponica* | 37 | TW | 40 | | 1 | 40 | 30 | 2 | 16 | 36 | |
| | 11 | *B. nipponica* | 8 | LY | 8 | | 3 | 8 | | | 6 | 8 | |
| | 52 | *B. nipponica* | 47 | MT | 51 | | | 51 | | | 22 | 47 | |
| | 31 | *B. nipponica* | 22 | FJ | 31 | | 5 | 31 | 22 | 4 | 22 | 22 | |
| | | *B. silvestriana* | 2 | | | | | | 2 | 2 | 2 | | |
| | 42 | *B. silvestriana* | 26 | GD | 34 | | 5 | 38 | 8 | 2 | 26 | | |
| | 14 | *B. silvestriana* | 3 | HK | 12 | 5 | | 10 | 2 | 2 | 3 | | |
| *F. formosana* | 35 | *B. silvestriana* | 22 | GD | 35 | | 5 | 34 | 2 | 2 | 22 | | |
| | 35 | *B. silvestriana* | 19 | HK | 35 | 8 | | 35 | 2 | 2 | 19 | | |
| | 61 | "*B. taiwanensis*" | 30 | TW | 51 | | 5 | 51 | 15 | 2 | 14 | 30 | |
| *F. vaccinioides* | 34 | *B. yeni* and *B. sp.* | 24 | TW | 26 | | 5 | 30 | 1 | 4 | 19 | 34 | |
| | 10 | *B. yeni* | 5 | LY | 10 | | 5 | 10 | 3 | | 2 | 5 | |
| *F. abelii* | 2 | *B. silvestriana* | 9 | GD | | 2 | 2 | | 2 | 4 | 9 | | |
| *F. pyriformis* | 14 | *B. silvestriana* | 16 | GD, HK | | 6 | 8 | | 4 | 4 | 16 | | |
| *F. variolosa* | 9 | *B. silvestriana* | 17 | GD, HK, FJ | | | 5 | | 6 | 5 | 17 | | |
| Outgroup (2 spp.) | 6 | | | | | | 6 | | | | | | |
| | | Outgroup (11 spp.) | 18 | | | | | | 12 | | 12 | | |
| Total | 528 | | 411 | | 418 | 21 | 60 | 447 | 134 | 37 | 329 | 263 | |

For abbreviations of population names, refer to Fig. 1; for detailed sample information, refer to Supplementary Data 1. The species names of figs and fig wasps are shown in italics.

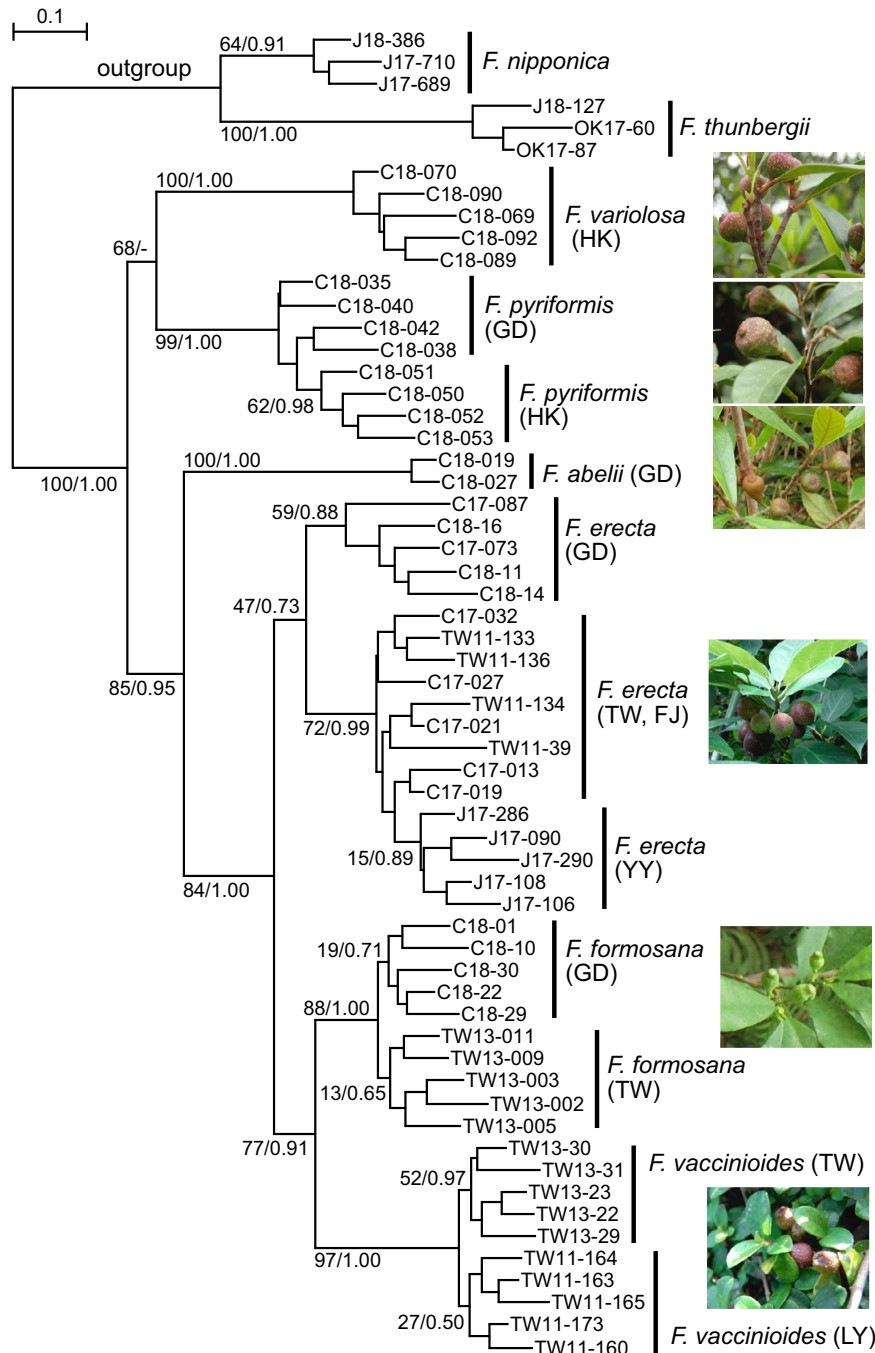

**Fig. 2 Phylogenetic tree of six closely related fig species based on genome-wide 2098 SNPs from 80138 loci of MIG-seq data.** *Ficus nipponica* and *F. thunbergii* were used as outgroups for the phylogenetic analysis. The numbers at the nodes show ML bootstrap values and BI posterior probabilities. For sample information, refer to Table 1 and Supplementary Data 1. For the abbreviations of population names in parentheses, refer to Fig. 1.

comprised the Guangdong (GD) population only and the other comprised the Fujian (FJ), Taiwan (TW) and Yaeyama Islands (YY) populations (Fig. 2). *Ficus formosana* and *F. vaccinioides* were divided into two phylogenetic groups, which corresponded to their geographical distributions (Fig. 2).

**Network of fig plastid DNA**. In plants, both seed and pollen dispersal promote gene flow between different populations. To confirm whether there is any differential dispersal of seeds and pollen, plastid DNA was analyzed from a total of 418 individuals belonging to populations of the three fig species, *F. erecta*, *F.*

*formosana*, and *F. vaccinioides* (Table 1). While analysis of the plastid DNA data did not yield detailed or robust phylogenetic relationships among the species and geographical populations analyzed due to the small sequence differences, haplotype analysis showed that there are 24 haplotypes and that these could be separated into three genetically distinct clusters (Supplementary Fig. 2). Cluster 2 and Cluster 3 were composed entirely of *F. erecta* samples from continental populations. Cluster 1 contained samples from all three species in all populations, except for the *F. erecta* samples from the FJ population. In other words, samples from the *F. erecta* complex distributed in the Taiwan-Ryukyu islands were grouped with *F. erecta* from Guangdong (GD) and

Hong Kong (HK). However, MIG-seq (Fig. 2) and microsatellite data (see the results in *Population genetic differentiation of figs* below) showed that the Taiwan-Ryukyu populations of *F. erecta* are more closely related to the FJ population than the GD and HK populations. This inconsistency suggests that there may be some dispersal of pollen, but not seeds, of *F. erecta* between the FJ and TW populations (Discussion).

**Analysis of ITS sequences of fig pollen carried by foundresses.** The pollinator wasps collected from the five fig species (*F. erecta*, *F. formosana*, *F. abelii*, *F. pyriformis* and *F. variolosa*) in southern China were all identified as belonging to the same species, *Blastophaga silvestriana*. To clarify whether there was any host specificity between *B. silvestriana* and the five fig species, we analyzed the ITS sequences of fig pollen carried by foundresses (i.e., wasps that enter receptive syconia for pollination or for laying eggs) and fig leaves (refer to the section, "PCR of plastid DNA and ITS sequences of figs" in Methods). Phylogenetic analysis showed that the ITS sequence of the pollen carried by a foundress (C18-040mA) collected from inside the receptive syconia of *F. pyriformis* was identical to that of *F. abelii*, but that it differed from *F. pyriformis* (Supplementary Fig. 3). This result suggests that the foundress collected from *F. pyriformis* carried pollen of *F. abelii*. In other words, pollinator wasps that are reared in *F. abelii* can enter the syconia of *F. pyriformis*, which implies that the pollinator wasps do not distinguish between the two host fig species. On the other hand, pollen carried by two foundresses (C18-074mA and C18-088mA) collected from inside the receptive syconia of *F. formosana* had identical ITS sequences to those of *F. formosana* (Supplementary Fig. 3).

**Phylogenetic relationships among pollinators.** Approximately 1900 bp of the mitochondrial (mt) *COI* and *COII* genes from 317 pollinator wasps of four *Blastophaga* species associated with six fig species (Table 1) were sequenced. A total of 203 different sequences were obtained and used for phylogenetic analysis with two outgroup species. The ML, BI and neighbour-joining (NJ) trees all showed the same topology, with all of the samples being grouped into five clades (Fig. 3 and Supplementary Figs. 4 and 5). Clade 1, which differs the most from the other clades (average *p*-difference: 10.6%), contained wasps collected from continental populations of the five fig species; all of these wasps were genetically indistinguishable and were morphologically identified as *B. silvestriana*. The pollinator wasps associated with *F. vaccinioides* were separated into Clade 2 and Clade 3, which correspond to the southern TW population (including the LY population) and the eastern TW population (Hualien), respectively (Supplementary Fig. 6). Clade 4 contained the pollinator wasps of two fig species, *F. erecta* (FJ, MT, TW and LY) and *F. formosana* (TW). Clade 5 contained the pollinator wasps from all of the Japanese populations of fig *F. erecta*. Interestingly, the pollinator wasps associated with *F. erecta* and *F. formosana* in the TW population were grouped into a single clade (Clade 4); together with the Japanese populations (Clade 5), the wasps in Clade 4 were split from one of the two clades of pollinator wasps associated with *F. vaccinioides* (Fig. 3). Since *B. taiwanensis*[38], which is described from the host fig *F. formosana*, is phylogenetically indistinguishable from *B. nipponica* (Fig. 3) and does not seem to be a distinct species, we have placed the name within double quotes throughout this paper.

The finding of genetically indistinguishable pollinators from different host fig species, as inferred by the *COI-COII* sequences, could be attributed to mitochondrial introgression. Therefore, to corroborate the results of the mtDNA sequence analyses, phylogenetic analysis was performed based on nuclear 28 S

rDNA and ITS1 sequences. Given the marked differences in the ITS1 sequences between *Blastophaga* and the outgroup species, these sequences could not be aligned correctly. As a result, phylogenetic analysis was performed separately for the 28 S rDNA sequences with outgroups, and the ITS1 sequences without outgroups. The pollinator wasps in Clade 1 and those in Clades 2–5 (Fig. 3) had identical 28 S rDNA sequences, respectively, and the difference between the two sequences (889 bp) was 0.8% (Supplementary Fig. 7). Phylogenetic analysis based on ITS1 sequences showed that there was a marked difference between *B. silvestriana* (Clade 1) and the other pollinator wasp species (Clades 2–5), but small sequence differences among Clades 2–5. Further, the phylogenetic tree shows that the *B. silvestriana* wasps from the five fig species were genetically indistinguishable (Supplementary Fig. 8). These results are essentially the same as those obtained from mtDNA sequences, except for the grouping of Clade 2 and Clade 3 (Supplementary Fig. 8).

**Species delimitation of *Blastophaga* wasps.** The phylogenetic clades of *Blastophaga* wasps did not correspond closely with the current classification of species (Fig. 3, Supplementary Figs. 4, 5, and 8). We used a Poisson Tree Processes (PTP) model to delimit species based on the phylogenetic tree obtained using *COI-COII* sequences. ML and Bayesian solutions gave the same delimitation results. Five species were identified as *Blastophaga* wasps with high support (Species 3–7 in Supplementary Fig. 9). Clades 1 and 3 were each assigned to a single species (i.e., Species 3 and 6), Clade 2 was separated into two species (Species 4 and 5), and Clades 4 and 5 were grouped into a single species (Species 7). These results suggested that the pollinator wasps associated with *F. vaccinioides* consist of more than one species and that *B. nipponica* and "*B. taiwanensis*" are probably the same species. Since Species 5 was composed of only one individual that was clearly grouped with individuals of Species 4, we tentatively treated Clade 2 as a single species, *B. yeni* (because *B. yeni* was described using the pollinator wasp individuals collected in the area corresponding to Clade 2)[38] and Clade 3 as *B.* sp., in this study.

**Divergence time of pollinators.** To estimate the divergence time of the pollinator clades shown in Fig. 3, we used *COI-COII* and 28 S rDNA sequences from pollinator wasps representing the clades shown in Fig. 3, another *Blastophaga* species (pollinator of *F. ischnopoda*), and ten species belonging to seven genera as outgroups. BEAST analysis (Supplementary Fig. 10) yielded the same phylogenetic topology obtained using mtDNA sequences alone (Fig. 3). The divergence times between clades were estimated as follows: Clade 1 and the others diverged ~5.12 million years ago (Ma), Clade 2 and Clades 3–5 diverged ~1.92 Ma, Clade 3 and Clades 4–5 diverged ~1.03 Ma, and Clades 4 and 5 diverged ~0.38 Ma (Supplementary Fig. 10).

**Population genetic differentiation of figs.** To investigate the genetic differentiation among the geographical populations of the three fig species (*F. erecta*, *F. formosana*, and *F. vaccinioides*), the population genetic structure was analyzed using nine microsatellite loci (SSR: simple sequence repeats) with a total of 447 individuals representing all of the populations of the three fig species (Table 1). When all three of the fig species samples were used, Evanno's $\Delta K$ showed a peak at $K = 2$, where individuals were assigned to two groups: Group 1 consisted of all individuals of *F. erecta*, except for the GD and HK populations; Group 2 included individuals of *F. formosana*, *F. vaccinioides* and the GD and HK populations of *F. erecta* (Supplementary Fig. 11, $K = 2$). We then performed separate analyses using samples from each

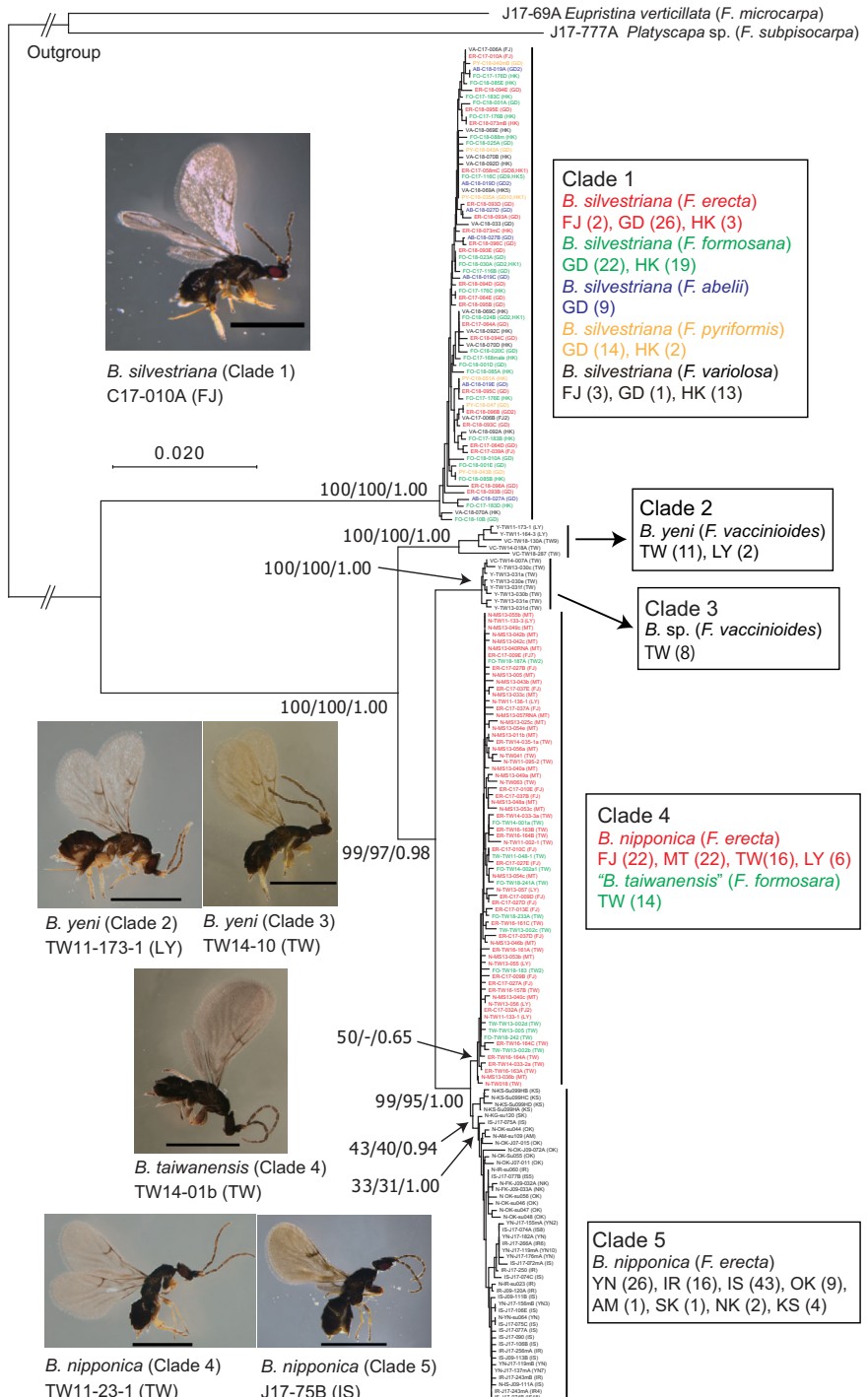

**Fig. 3 NJ tree of pollinator wasps based on *COI-COII* sequences.** Two species, *Eupristina verticillata* (host fig: *F. microcarpa*) and *Platyscapa* sp. (host fig: *F. subpisocarpa*) were used as outgroups. There were a total of 1844 sites in the final dataset after eliminating gaps and missing data. The ML and BI analyses showed the same tree topology among the clades (Supplementary Figs. 4 and 5). This tree topology also matches that based on nuclear 28 S rDNA and ITS1, except for the grouping of Clade 2 and Clade 3 in the ITS1 tree (see Supplementary Figs. 7 and 8). The numbers at the nodes show NJ and ML bootstrap values and BI posterior probabilities (NJ/ML/BI). The population names of samples are given in parentheses after sample names, and the number after the population name indicates the number of individuals with the same sequences (see Supplementary Data 1). Species names of wasps and their host figs (in parentheses), and population name and number of individuals (in parentheses) are shown for each clade on the right side of the tree. For the wasp pictures, the species name (clade number), individual number (population name) and the scale bars (1 mm) are shown below each picture. For the abbreviations of population names, refer to Fig. 1.

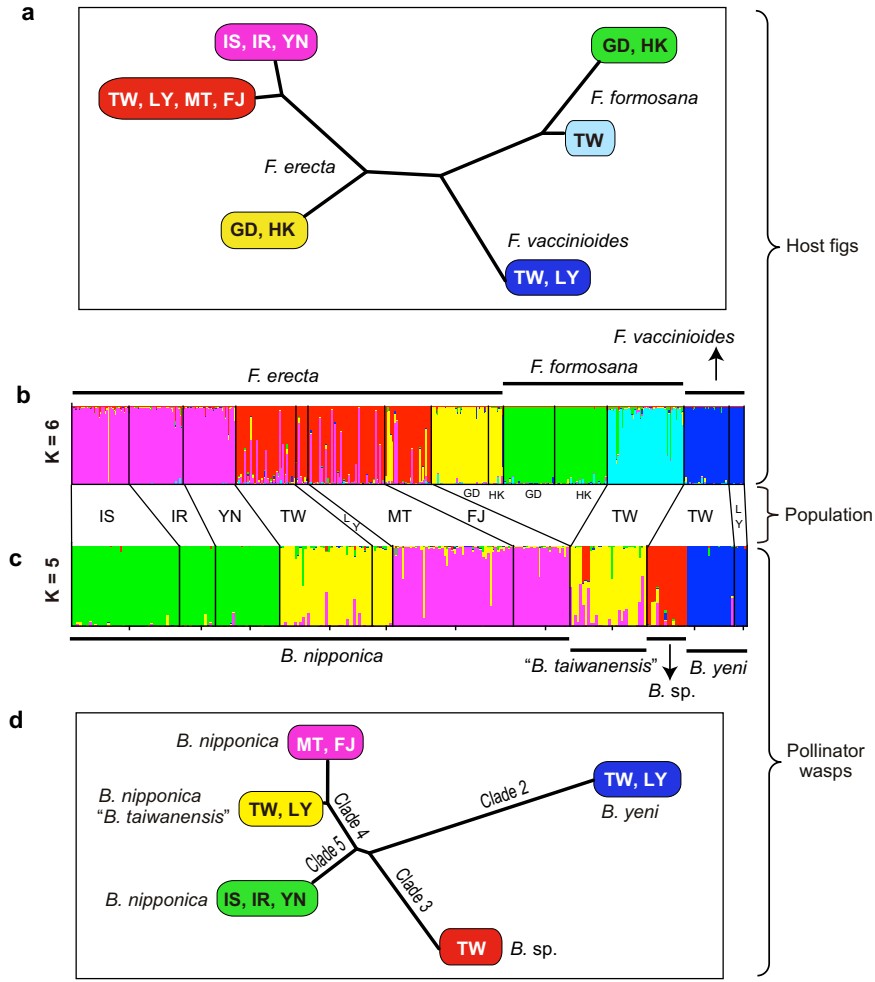

**Fig. 4 Population genetic structures (b and c) and trees (a and d) of the clusters based on SSR data.** SSR data were genotyped from nine loci for figs and from 15 loci for pollinator wasps. $K = 6$ and $K = 5$ were the most likely $K$ for figs and wasps, respectively (structures with $K = 5$ and $K = 7$ are also shown for figs in Supplementary Fig. 11). The trees were computed by applying the NJ algorithm to the matrix of allele-frequency divergence among clusters[69,70]. The samples used in these analyses are shown in Table 1 and Supplementary Data 1. For the abbreviation of population names, refer to Fig. 1. The clades shown in (**d**) correspond to those in Fig. 3.

group, and the Evanno's $\Delta K$ showed peaks at $K = 2$ for Group 1 and at $K = 4$ for Group 2, with the individuals from Group 1 and Group 2 classified into two and four separate clusters, respectively (Fig. 4b). These genetically distinct clusters correspond to the following populations: IS/IR/YN (Yaeyama populations), TW/LY/MT/FJ and GD/HK of *F. erecta*, GD/HK and TW of *F. formosana*, and TW/LY of *F. vaccinioides* (Fig. 4b). When $K \geq 3$ for Group 1, individuals from TW/LY/MT/FJ were further classified into two distinct clusters (TW/LY and MT/FJ), although some individuals from TW were genetically similar to individuals from MT and FJ (Supplementary Fig. 11, $K = 7$). The phylogenetic relationships among the genetic clusters ($K = 6$) showed that the GD population of *F. erecta* was largely separated from other *F. erecta* populations, which were closely related to each other (Fig. 4a). In addition, the two clusters of *F. formosana* were also closely related to each other (Fig. 4a and Supplementary Fig. 11, $K = 5$).

Pairwise $F_{ST}$-values between the populations of different species ranged from 0.2 to 0.34, suggesting that there are very large genetic differences among the three fig species (i.e., *F. erecta*, *F. formosana*, and *F. vaccinioides*) (Supplementary Table 1). Pairwise $F_{ST}$-values between the GD population and the other *F. erecta* populations ranged from 0.13 to 0.21, suggesting extensive genetic differentiation. Moderate genetic differentiation was

observed between the IS/IR/YN and TW/MT/FJ populations ($F_{ST}$-values: 0.08–0.10), between the TW and MT/FJ populations ($F_{ST}$-values: 0.05–0.06) of *F. erecta*, and between the TW and GD/HK populations of *F. formosana* ($F_{ST}$-values: 0.12–0.14). Some genetic differentiation was observed between the YN and IS/IR populations ($F_{ST}$-values: 0.02–0.025) and between the FJ and MT populations ($F_{ST}$-value: 0.025) (Supplementary Table 1).

**Population genetic differentiation of pollinators**. Genetic structure analysis was performed based on the SSR data (15 loci) from a total of 263 individuals representing all populations of the pollinators *B. nipponica*, "*B. taiwanensis*", *B. yeni*, and *B. sp.* (Table 1). Since *B. silvestriana* differed markedly from the other wasp species included in the phylogenetic analysis (Fig. 3), the samples of this species were excluded from the analysis of population structure. The results showed a distinct peak for $\Delta K$ at $K = 5$, where individuals were assigned to five different genetic clusters (Fig. 4c). Interestingly, the individuals of *B. nipponica* and "*B. taiwanensis*" in the TW population were assigned to the same cluster (yellow in Fig. 4c), but *B. sp.* and *B. yeni* formed distinct clusters (red and blue in Fig. 4c), which corresponded with the eastern TW population (Hualien) and the southern TW population (including the LY population), respectively (Supplementary Fig. 6).

The phylogenetic relationships among these genetic clusters showed that *B. yeni* and *B*. sp. are markedly different from each other and also from the other clusters (Fig. 4d), although they are associated with the same host species, *F. vaccinioides*. The three Ryukyu populations (IS, IR and YN) were clearly separated from other populations of *B. nipponica* and "*B. taiwanensis*" (Fig. 4d). These results are consistent with the findings of the phylogenetic analysis based on mtDNA and ITS1 sequences (Fig. 3 and Supplementary Fig. 8).

In the pairwise $F_{ST}$ analysis (Supplementary Table 2), considerable genetic differentiation was observed between *B. yeni*/*B.* sp. (TW) and the other populations. Two Ryukyu populations of *B. nipponica* (IS and YN) also differed markedly from the other populations. Moderate levels of genetic differentiation were observed between the TW and MT/FJ populations of *B. nipponica*, and a slight, but significant, genetic difference was also observed between *B. nipponica* and "*B. taiwanensis*" in the TW population (Supplementary Table 2). These results also corroborated those obtained from the population structure analysis (Fig. 4c and d) and mtDNA sequence-based phylogenetic analysis (Fig. 3).

**Cophylogenetic analysis between figs and pollinators**. To determine whether the fig-pollinator pairs coevolved with each other, a software programme for phylogenetic tree reconciliation under the duplication-transfer-loss model was used to compare the phylogenetic relationships of figs and pollinators (Supplementary Fig. 12a). The results showed that a total of 11 events, including five cospeciation events, five transfers and one loss, have occurred among the figs and pollinators over evolutionary time; of these events, seven were supported robustly (Supplementary Fig. 12b). As mentioned in the "Methods" section, since the *B. silvestriana* wasps from the five host fig species were genetically indistinguishable (Clade 1 in Fig. 3), the inferred events (i.e., three transfers and one cospeciation, Supplementary Fig. 12b) between the five fig species and *B. silvestriana* wasps are not discussed here.

## Discussion

The phylogenetic relationships among the figs and pollinator wasps estimated in this study are summarized in Fig. 5. The six fig species (*F. variolosa*, *F. pyriformis*, *F. abelii*, *F. erecta*, *F. formosana*, and *F. vaccinioides*) are all closely related, but their morphologies are extremely variable[37]. The molecular analyses based on MIG-seq and SSR data showed that these fig species were clustered into distinct groups (Figs. 2 and 4a). *Ficus erecta* and *F. formosana* were divided into several genetically and geographically distinct populations (Fig. 4a and b). In contrast to the host figs, the pollinator wasps collected from the five fig species distributed in southern China formed a single clade (Clade 1 in Fig. 3) in which no genetic differentiation was detected. However, the pollinator wasps from the three fig species found in Taiwan formed four different clades that were clearly distinct from Clade 1 (Fig. 3). The *COI-COII* sequence differences within Clade 1 (Fig. 3) were very small (pairwise *p*-distance < 0.8%). The range of sequence divergence in the *COI* gene at the species level is typically considered to be $0.8 \pm 0.6\%$[35] and less than 1.4%[23]. Compared to these previous findings, the sequence differences within Clade 1 were considered to be at the intraspecific level. The phylogenetic analysis based on nuclear ITS1 sequences corroborated the results obtained from the *COI-COII* sequence analyses (Supplementary Fig. 8), suggesting that the small differences in *COI-COII* sequences of the pollinator wasps from the five host fig species were not due to mitochondrial introgression. Species delimitation based on phylogenetic trees suggests that Clade 1

consists of only one species (Supplementary Fig. 9), and all of the wasps in Clade 1 were morphologically identified as *Blastophaga silvestriana*[39]. Thus, our molecular results combined with morphological observations strongly suggest that the five dioecious fig species, which are both molecularly (Figs. 2 and 4a) and morphologically[40] distinct from each other, share a single pollinator species, *B. silvestriana*, in southern China (Fig. 5). Although pollinator sharing has been reported previously in five dioecious fig taxa[25], each of those fig species has its own dominant pollinator which is shared with one or two other fig species. To the best of our knowledge, the present study is the first to describe a case in which so many distinct dioecious fig species are associated with a single pollinator species. In addition, examples of pollinator sharing have been mainly reported in monoecious figs, and the incidence among dioecious figs is considered to be low[23,35]. However, the findings of the present study suggest that pollinator sharing events occur more frequently in dioecious fig species than was previously thought, as suggested in two previous studies[24,25].

On the other hand, it is unclear how these host fig species maintain their unique species characteristics when sharing pollinators. One possibility is that the pollinator wasps from the different fig species are biologically distinct species that are genetically and morphologically indistinguishable. The wasp species may be able to discriminate among host figs by detecting floral scents emitted from the syconia of fig plants. However, analyses based on ITS sequences of fig pollen carried by foundress wasps show that female wasps reared in *F. abelii* can enter the receptive syconia of *F. pyriformis* (Supplementary Fig. 3). Although our current data are limited, these findings imply that the pollinator wasps may not be able to distinguish between these two fig species. Another possibility is that fig species avoid interspecies hybridization through differences in phenology, pollen incompatibility, or hybrid sterility[41]. Further studies on the pollination behaviour of pollinator wasps, phenological observations, and hybridization experiments of figs are therefore required in order to clarify these possibilities.

As mentioned in the Introduction, numerous examples of co-pollination have been reported, not only in monoecious figs but also in dioecious figs[23,24]. Most of these examples were identified based on phylogenetic analyses of pollinator wasps alone[17], with the exception of a few recent studies in which SSR data showed genetic differences among closely related fig species and populations[24–26]. In the present study, we estimated the phylogenetic relationships among six closely related fig species and some of their constituent populations using analyses of genome-wide sequence data and SSR data. These analyses enabled us to more precisely compare the phylogenetic relationships between these fig species and their pollinator wasps (Fig. 5).

Through the comparisons, we identified three examples of co-pollination. First, the Fujian (FJ) population of *F. erecta* is associated with two sympatric pollinator species, namely *B. nipponica* and *B. silvestriana*. These two pollinator species were even collected from the same fig tree (albeit, from different syconia) (Supplementary Data 1 and Fig. 3). This example of co-pollination was most likely due to dispersal of *B. nipponica* from Taiwan (Figs. 5 and 6d), and the dispersal seems to be unidirectional, namely from Taiwan to the Fujian region. The possible reasons for the unidirectional dispersal of *B. nipponica* are as follows: (1) *B. nipponica* diverged from the pollinator lineage, *B*. sp. (Fig. 3) or the common ancestor of *B*. sp. and *B. yeni* (Supplementary Fig. 8), which are endemic to Taiwan; (2) *B. nipponica* was previously only known to occur in Taiwan and Japan[12], so this is the first record of *B. nipponica* collected from continental China; (3) From the sampling conducted as part of this study, *B. nipponica* is not widely distributed in continental China, and seems to be limited to the Fujian region. In addition, if

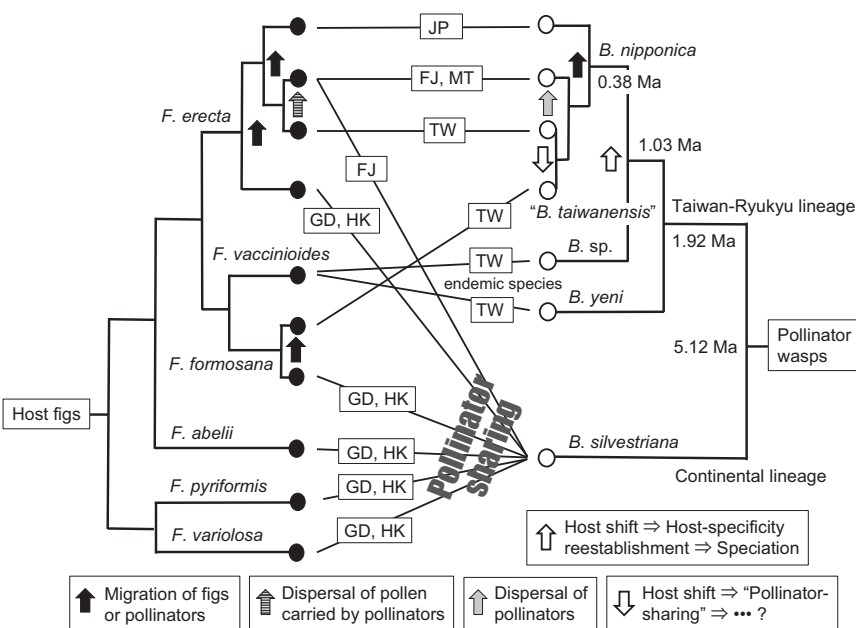

**Fig. 5 Summary of phylogenetic relationships of figs and their pollinator wasps.** The fig tree topology is based on the results obtained from MIG-seq and SSR data (Figs. 2 and 4a), while that of pollinators is based on the results inferred from *COI-COII* sequences and SSR data (Figs. 3 and 4d). The possible evolutionary events, including migrations of figs and pollinators, pollen and wasp dispersal, and host shifting are indicated (for host shifting events, see Supplementary Fig. 12b). Solid circles and open circles show figs and pollinators, respectively. The divergence times between pollinator lineages are from the results of the dating analysis (Supplementary Fig. 10). JP Japan, FJ Fujian, MT Matsu Islands, TW Taiwan, GD Guangdong, HK Hong Kong.

the dispersal of pollinator wasps was bidirectional, then it would be expected that the continental species, *B. silvestriana*, would occur in Taiwan. However, there are no records of *B. silvestriana* in Taiwan or Japan[12], and we have never collected individuals of *B. silvestriana* from either of these regions; however, a failure to collect specimens is not equivalent to absence. It is possible that unidirectional dispersal may have been caused by typhoons because fig-pollinating wasps are small insects (~2 mm in body length) and are unable to disperse long distances under their own power. They actively fly above the canopy and then rely on the wind to disperse them passively over long distances[42,43]. On average, about four typhoons a year pass through Taiwan and reach the Fujian region of Mainland China (Central Weather Bureau, Taiwan), and it seems highly likely that these typhoons transport pollinator wasps from Taiwan to the Fujian region.

Second, the pollinator of *F. vaccinioides*, is divided into two genetically distinct clades (Clade 2 and Clade 3 in Figs. 3, 4c and d), and the difference in sequences between the clades seems to be at the species level (*B. yeni* and *B.* sp. in Supplementary Fig. 9). Since no morphological differences were observed between the two clades, they were regarded as two phylogenetic (or cryptic) species that duplicated within the host fig, *F. vaccinioides*, probably due to geographical diversification[13] ~1.92 Ma (Fig. 5). *Ficus vaccinioides* has a vine life-form and grows on the ground. These ecological characteristics may limit the dispersal of its pollinator(s) and promote their geographic divergence. However, no genetic differentiation has been observed between the host figs of these two clades (Fig. 4a and b). Thus, it seems likely that *F. vaccinioides* is associated with two pollinator species, but this is not considered to be a case of cospeciation as inferred by cophylogenetic analyses (Supplementary Fig. 12b), as these are based on tree topologies without genetic distances.

Third, *F. formosana* is associated with two allopatrically distributed pollinators, "*B. taiwanensis*" in Taiwan and *B. silvestriana* in continental China. This case of co-pollination is attributed to host shifting, with *B. nipponica* shifting from its original host, *F. erecta*, to *F. formosana* (Supplementary Fig. 12b, Figs. 5, 6c). It is

considered that the occurrence of co-pollination in dioecious figs is solely due to duplication (i.e., co-pollinator species are in sister relationships) of pollinator wasps within the same host species[17,23]. However, our findings showed that co-pollination events can be caused by host shifting in closely related dioecious figs.

Given the obligate mutualistic relationships between figs and their pollinator wasps[7,9], long-distance island colonization of fig species is limited, due to the difficulties associated with synchronous co-dispersal with their pollinators[44,45]. However, our results revealed that colonization of species in the *F. erecta* complex[37] has occurred repeatedly from continental China to Taiwan, implying that the key to successful colonization may be the reestablishment of species-specific relationships with new pollinators through host shifting (Figs. 5 and 6b).

The origin of the *F. erecta* complex and its pollinators in the Taiwan-Ryukyu islands begins with the establishment of *F. vaccinioides* and its pollinators (*B. yeni* and *B.* sp.), which were co-split from the *F. formosana* and *B. silvestriana* lineage, respectively (Figs. 2 and 3 and Supplementary Fig. 12b). Both *F. vaccinioides* and its pollinators are strictly endemic to Taiwan. *Blastophaga yeni* and *B.* sp. are the phylogenetically oldest lineages among the pollinators of the *F. erecta* complex in the Taiwan-Ryukyu islands (Fig. 3), and their common ancestor was estimated to have diverged from *B. silvestriana* (the continental lineage of *Blastophaga*) ~5.12 Ma (Supplementary Fig. 10). The Taiwan-Ryukyu islands are considered to have emerged about 9 Ma and attained their modern features and their current flora and fauna from the adjacent continent only 5–6 Ma[46]. Based on these findings, it is reasonable to assume that the ancestors of *F. vaccinioides* and its pollinators (*B. yeni* and *B.* sp.) inhabited the ancient Taiwan region on the continent, and these taxa became established after Taiwan split from the continent (Fig. 6a). Of course, it is also possible that *F. vaccinioides* and its pollinators may have migrated to Taiwan from continental China after Taiwan split away from the continent, but we have not yet identified continental pollinator wasps that are more closely related to *B. yeni* or *B.* sp. than to *B. silvestriana*.

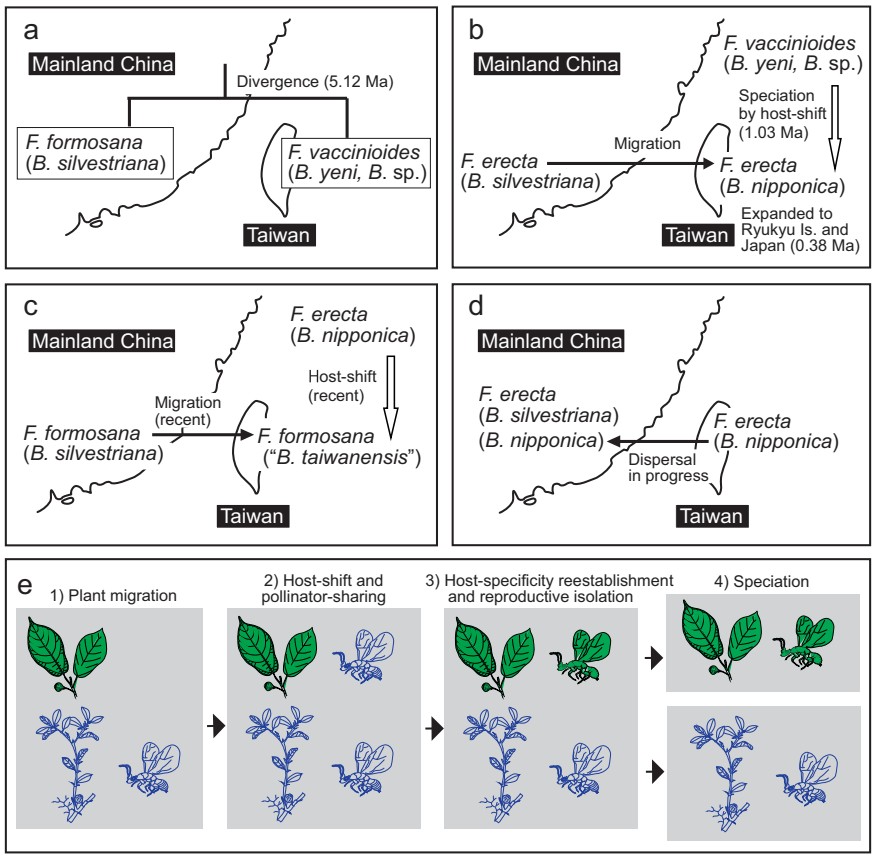

**Fig. 6 Establishment of the mutualistic relationships in the *F. erecta* complex and its pollinators (in parentheses) in the Taiwan-Ryukyu islands (a–d) and a model for host-shift speciation proposed in this study (e). a** The divergence of *F. vaccinioides–B. yeni/B.* sp. and *F. formosana–B. silvestriana.* **b** *F. erecta* migrated into Taiwan from mainland China. *Blastophaga* sp. host-shifted to *F. erecta* and speciated into *B. nipponica.* Then, the distribution range of *F. erecta–B. nipponica* expanded northward to the Ryukyu Islands and Japan. The chronology in **a** and **b** is based on the dating analysis of the pollinators (Supplementary Fig. 10). **c** *F. formosana* probably migrated into Taiwan recently from mainland China, and *B. nipponica* host-shifted to *F. formosana*; at present, *F. erecta* and *F. formosana* appear to share *B. nipponica* including "*B. taiwanensis*" in Taiwan. **d** *B. nipponica* dispersed from Taiwan to the Fujian region (FJ) of mainland China, resulting in an example of sympatric co-pollination. **e** The proposed model for the host-shift speciation events described in this study consists of four stages: (1) Host figs migrate into a new niche without their original pollinators; (2) The migrated host figs are pollinated by new wasps (host shifting); as a result, two host fig species share one wasp pollinator (pollinator sharing); (3) The host-shifted wasps evolve and become associated with the host figs that migrated (host-specificity reestablishment), and reproductive isolation occurs; (4) Genetic divergence proceeds and speciation occurs. These images show only the evolutionary events, not the details of the evolution of figs and pollinators during these processes.

Considering the phylogenetic relationships among the members of the *F. erecta* complex (Fig. 2), *F. erecta* was probably established somewhere in continental China before the divergence of *F. vaccinioides* and *F. formosana*. Based on the genetic divergence between fig populations (see pairwise $F_{ST}$ data in Supplementary Table 1) and the divergence times estimated for the pollinator wasp lineages (Supplementary Fig. 10), *F. erecta* colonized Taiwan after the divergence of *F. vaccinioides* and *F. formosana* and acquired a new pollinator (*B. nipponica*) through host shifting of the pollinators associated with *F. vaccinioides* (Supplementary Fig. 12b) about 1.03 Ma. Later, during the late quaternary glacial period at around 0.38 Ma, the distribution of *F. erecta* and its pollinator (*B. nipponica*) extended to the Ryukyu Islands and the Kanto region of Japan, currently the northernmost boundary of the genus *Ficus* (Figs. 5 and 6b).

Given the similarities in the plastid DNA of *F. erecta* between the GD/HK populations and the populations of the Taiwan-Ryukyu islands (Supplementary Fig. 2), *F. erecta* probably colonized Taiwan from the GD/HK region rather than from the FJ region. However, the SSR data imply that the TW and FJ populations of *F. erecta* are most closely related (Fig. 4 and Supplementary Tables 1 and 2), which is inconsistent with the results

obtained from the plastid DNA (Supplementary Fig. 2). Therefore, the most reasonable explanation would be those pollinator wasps (*B. nipponica*) that were carrying pollen dispersed from Taiwan to the Fujian area, as discussed above, resulting in hybridization of the figs between the two areas through the dispersal of pollen, but not seeds. As a result, the nuclear DNA of the FJ population became more similar to that of the TW population (Fig. 4a, b, and Supplementary Table 1), while plastid DNA findings show a more distant relationship with the FJ population (Supplementary Fig. 2). This explanation seems to be supported by the population genetic analysis, in which some individuals of the FJ population have a genetic structure that is similar to that of the GD population (yellow in Fig. 4b). In contrast to the host figs, there is no evidence to suggest that hybridization occurred between the two pollinator species, *B. nipponica* and *B. silvestriana*, although they are sympatrically associated with the same host fig species (*F. erecta*) in the FJ region.

For *F. formosana*, a slight (but significant) genetic difference was detected between the TW and GD/HK populations (Supplementary Table 1), implying that this fig species colonized Taiwan very recently (Figs. 5 and 6c). *Ficus formosana* likely acquired its pollinator, "*B. taiwanensis*", through host-shifting of

*B. nipponica* (Supplementary Fig. 12b, Figs. 5 and 6c) (see the discussion below).

The maintenance of interactions between organisms in mutualistic relationships is often difficult to reconcile with natural selection[47,48]. Specialized mutualism may be lost when plants colonize islands in the absence of their mutualistic partners; forcing the plant to switch from a specialized mode of pollination to a more generalized mode of pollination[33]. However, our findings suggest that fig species appear to be capable of recruiting and establishing species-specific relationships with new pollinators, enabling them to successfully colonize suitable habitats.

Fig-wasp mutualism is a classic example of cospeciation, but there is little direct evidence to support strict cospeciation; indeed, numerous molecular studies show phylogenetic incongruence between closely related figs and their pollinator wasps[8,13,23]. Moreover, it has been suggested that pollinator wasps may undergo speciation by host shifting or by duplication, instead of by strict cospeciation[13]. Our current findings clarified the detailed mechanisms underlying host-shift speciation; specifically, the speciation of the pollinator *B. nipponica*. Host-shift speciation comprises four stages (Fig. 6e): (1) The host fig, *F. erecta*, migrated to Taiwan from mainland China; (2) some individuals of the pollinator wasp *B.* sp., host-shifted to *F. erecta* from their original host *F. vaccinioides*; (3) these pollinator wasps then became specialized and became associated with *F. erecta*; and (4) the pollinators then differentiated from *B.* sp. and speciated to become *B. nipponica* (Fig. 6b and e). Of course, host figs may have also evolved over the course of these evolutionary processes; for example, the evolution of the vine life-form in *F. vaccinioides*.

Similarly, *F. formosana* also migrated to Taiwan from mainland China, but later than *F. erecta* (Fig. 5). This was followed by host-shifting of *B. nipponica* from *F. erecta* to *F. formosana* (Supplementary Fig. 12b, Figs. 5 and 6c). The pollinator associated with the Taiwanese *F. formosana* has been morphologically described as "*B. taiwanensis*"[38]. However, "*B. taiwanensis*" does not appear to be a distinct species from *B. nipponica* because they are phylogenetically indistinguishable (Clade 4 in Figs. 3 and 4c). It is possible that the two fig species, *F. erecta* and *F. formosana*, share the pollinator *B. nipponica*/"*B. taiwanensis*" in the TW population. Indeed, network analysis of plastid DNA showed that several *F. erecta* individuals have the same haplotype sequence as *F. formosana* (h9 in Supplementary Fig. 2), suggesting that these individuals originated from a hybrid between *F. erecta* and *F. formosana*. However, no evidence of hybridization between the two species was observed based on morphological observations and population genetic analysis of SSR data (Fig. 4b). Hybridization between the two fig species may therefore have been limited to shortly after *F. formosana* migrated to Taiwan and is no longer occurring. Supporting evidence was obtained from the analysis of SSR data, which showed that there was significant population genetic differentiation between the pollinators *B. nipponica* and "*B. taiwanensis*" (Supplementary Table 2). These findings imply that the host-specificity of "*B. taiwanensis*" may have proceeded to some extent, leading to reproductive isolation between *B. nipponica* and "*B. taiwanensis*" and preventing hybridization between their hosts (*F. erecta* and *F. formosana*) by restricting pollen flow; however, as discussed above, there are other factors that prevent hybridization between host figs. Consequently, testing whether the species-specificity between *F. formosana* and "*B. taiwanensis*" has been established will be an important future challenge. Such analysis will provide more direct evidence for whether "*B. taiwanensis*" is in the process of speciation.

The findings of the present study show that under certain circumstances, the species-specific relationships between figs and pollinators can be disrupted and reestablished through the acquisition of new pollinators through host shifting. Such host shifting could potentially lead to co-pollination or pollinator sharing, and pollinator speciation by reestablishment of host-specificity. A recent study[49] presented genomic evidence of prevalent hybridization between fig species, implying that host-shifting events might have occurred frequently over the evolutionary history of fig-wasp pollination mutualism. Our study provides important findings for understanding the outcomes of pollinator host-shifting between fig species. After host-shifting, pollinator sharing may continue for some time between the two host species. Two scenarios are considered for pollinator sharing[24]. One is that pollinator sharing is maintained, increasing the chances of hybridization between the two host fig species. The other scenario is that the host-shifted pollinator wasps adapt to become associated with the new host species, leading to genetic differentiation and pollinator speciation. The findings of the present study provide evidence to support the second scenario, and suggest that fig-wasp pollination mutualism may exert an evolutionary force or selection pressure through factors such as differences in floral scents and/or syconium morphology, that may facilitate the establishment of host-specificity in pollinators. Such an evolutionary force may be central to understanding the maintenance of host-pollinator specificity and the speciation mechanisms found in fig-wasp pollination mutualism.

## Methods

**Sampling**. Six dioecious fig species (*F. erecta*, *F. formosana*, *F. vaccinioides*, *F. abelii*, *F. pyriformis*, and *F. variolosa*) and their pollinator wasp species were examined in this study. As mentioned in the Introduction, these fig species were considered to be well suited for this study because they are distributed in the Taiwan-Ryukyu islands and/or the adjacent continent (mainland China), and are most closely related with each other in the *F. erecta* complex, which consists of 17 species[37]. According to the Flora of China (www.eFloras.org), the approximate distributions of these figs and their pollinator wasps are as follows: *F. erecta*: S. China, Taiwan, Ryukyu Islands and W. Japan, Vietnam, and Cheju Island (Korea); *F. formosana*: S. China, Taiwan, and N. Vietnam; *F. vaccinioides*: endemic to S. Taiwan; *F. abelii*: S. China, part of S. Asia, and part of S.E. Asia; *F. pyriformis*: S. China and Vietnam; *F. variolosa*: S. China, Laos and Vietnam. We sampled these fig species and their pollinator wasps from S. China (Fujian and Guangdong), Hong Kong, Taiwan, Lanyu Island, Matsu Islands, Ryukyu Islands, and W. Japan (Fig. 1). Detailed sample information is listed in Supplementary Data 1.

For sampling of figs, fresh leaves were collected from fig trees, dried using silica gel in the field, and kept in the laboratory until DNA extraction. For sampling of pollinator wasps, mature, but unopened, syconia (one to five per tree) were collected from male fig trees and kept in a 30 ml plastic tube for collection of wasps. Pollinator wasps that emerged from the syconia were preserved in 70–80% and 99.5% ethanol solution for morphological examination and DNA analyses, respectively. Some pollinator wasps were collected from inside the receptive syconia of fig trees. These wasps are generally called foundresses, which entered the receptive syconia for pollination and/or to lay eggs. To avoid using pollinator wasps from close relatives, in this study, we used wasp samples from syconia that were collected from different fig trees where possible. The sample numbers of figs and pollinator wasps collected from each population of each species are shown in Table 1, and their details are shown in Supplementary Data 1.

Since the pollinator wasps reared from the five fig species (*F. erecta*, *F. formosana*, *F. abelii*, *F. pyriformis* and *F. variolosa*) in S. China and Hong Kong were morphologically indistinguishable, we treated them as *Blastophaga silvestriana* according to Grandi[50] and Hill[39]. In the Fujian population, two pollinator wasp species, *B. silvestriana* and *B. nipponica*, which are morphologically distinguishable, were collected from one fig species, *F. erecta*; even from the same individual fig tree. In the Taiwan population, it is difficult to discriminate between the two pollinator species associated with *F. erecta* and *F. formosana* by morphology. We therefore tentatively treated the pollinator wasps from *F. erecta* as *B. nipponica*, and those from *F. formosana* as "*B. taiwanensis*"[38].

We collected leaf samples from a total of 522 fig trees belonging to six fig species, i.e., 322 from *F. erecta*, 131 from *F. formosana*, 44 from *F. vaccinioides*, two from *F. abelii*, 14 from *F. pyriformis*, and 9 from *F. variolosa*. In addition, pollinator wasp samples were collected from the fig trees, i.e., 251 from *F. erecta*, 71 from *F. formosana*, 29 from *F. vaccinioides*, nine from *F. abelii*, 16 from *F. pyriformis*, and 17 from *F. variolosa* (Table 1 and Supplementary Data 1). Specimens of figs and pollinator wasps are deposited at the Biohistory Research Hall (BRH), Osaka, Japan.

**DNA extraction from figs**. Total genomic DNA was extracted from the obtained leaf materials (about 10 mm$^2$) dried with silica gel using a DNeasy Plant Mini Kit (Qiagen) according to the manufacturer's instructions. The DNA samples were finally eluted in a 200 µL AE buffer (Qiagen). These DNA samples were used as templates for PCR amplification of plastid DNA, ITS, MIG-seq, and SSR sequences (Table 1 and Supplementary Data 1).

**DNA extraction from pollinator wasps**. Total genomic DNA was extracted from whole-body specimens of wasp samples preserved in ethanol using a DNeasy Blood and Tissue Kit (Qiagen) according to the manufacturer's instructions. The DNA samples were finally eluted in 100 µL AE buffer (Qiagen) and used as a template for PCR amplification of the nuclear 28 S rDNA, ITS1, *COI-COII*, and SSR sequences (Table 1 and Supplementary Data 1).

**MIG-seq experiment for figs**. To investigate the phylogenetic relationships among the six closely related fig species, genome-wide SNPs detected by the MIG-seq technique[51] were used for phylogenetic analysis. A total of 60 fig samples representing the eight fig species (including two outgroup species) were used for the MIG-seq analysis (Table 1 and Supplementary Data 1). The two outgroup species, *F. nipponica* and *F. thunbergii* were selected from the subgenus *Synoecia*, which is most closely related to the *Ficus* section to which the *F. erecta* complex belongs[28]. MIG-seq libraries were prepared following the method of Suyama and Matsuki[51], with slight modifications. Specifically, the annealing temperature for the first PCR was changed to 38 °C and the number of cycles used for the second PCR was changed to 20 cycles. The PCR fragments (libraries) were quantified on a Synergy H1 plate reader (BioTek) using QuantiFluor dsDNA System (Promega), and the quality of the product was determined on an Agilent 2100 bioanalyzer using a High Sensitivity DNA kit (Agilent Technologies). The libraries from each sample were pooled in equimolar concentrations. The mixed libraries were then purified and fragments in the size range of 200–1000 bp in the purified library were isolated using AMPure XP beads (Beckman Coulter). The final PCR product libraries were sequenced (2 × 76 bp) on an Illumina NextSeq 500 system using a NextSeq 500/550 High Output Kit v2.5 (Illumina).

**SNP detection**. The adaptor and primer were removed in the sequenced reads using FASTX trimmer (http://hannonlab.cshl.edu/fastx_toolkit/). Low-quality nucleotide sites (quality score under 30) and low-quality reads (sequence length under 50 bp) were removed from raw reads using Sickle v1.33 (https://github.com/najoshi/sickle). For phylogenetic analyses, the cleaned reads were unified to 50 bp in length. De novo assembly and SNP detection were conducted using Stacks v2.2[52]. The minimum depth option for creating a stack was set to 5, and default settings for all other options were used. Stacks v2.2 was also used to select phylogenetically informative SNP sites for phylogenetic analysis by removing sites in which there were no SNP differences between samples, such as single SNPs.

**PCR of plastid DNA and ITS sequences of figs**. Plastid DNA, including six non-coding regions, the *rps16* intron, *trnG* intron, *petB* intron, *trnL* intron, *trnL-trnF* spacer region, and the *atpB-rbcL* spacer region were amplified from a total of 418 samples of three fig species (i.e., *F. erecta*, *F. formosana*, and *F. vaccinioides*) (Table 1 and Supplementary Data 1).

Since the pollinator species *B. silvestriana* appeared to be associated with five distinct fig species in S. China and Hong Kong, we sought to investigate whether there was any host specificity between *B. silvestriana* and the five fig species. We specifically examined whether pollinator wasps could enter the receptive syconia of a fig species different from the species in which they were reared. We did this by comparing the ITS sequences of fig leaves and pollen carried by foundresses. The ITS sequences were amplified from 18 leaf DNA samples of four fig species (*F. erecta*, *F. formosana*, *F. pyriformis*, and *F. abelii*), and pollen DNA samples from three foundresses (one from *F. pyriformis* and two from *F. formosana*) (Supplementary Fig. 3).

PCR amplification was performed using LA-Taq (TaKaRa) or KOD-Plus (TOYOBO) DNA polymerase in 25 µL reaction mixtures containing 1 µL (figs) or 2 µL (pollinator wasps) of template DNA and following the procedures outlined in previous studies[16,24,32,53] with some modifications. The PCR primers and conditions are shown in Supplementary Tables 3 and 4, respectively.

**PCR of 28S rDNA, ITS1, and *COI-COII* sequences in pollinators**. These sequences were used for phylogenetic analyses of pollinator wasps. The *COI-COII* sequences of a total of 317 wasp individuals of four *Blastophaga* species were amplified together with 12 outgroup wasps belonging to 11 species (Table 1). To confirm the results obtained from the *COI-COII* sequence data, nuclear 28 S rDNA and ITS1 were amplified from some wasp samples (122 samples including 12 outgroups for 28S rDNA analysis and 37 samples for ITS1 analysis) (Table 1). PCR amplification of these DNA sequences was performed using the primers shown in Supplementary Table 3 and the conditions shown in Supplementary Table 4.

**Sequencing of PCR products**. The PCR products were enzymatically cleaned with Calf Intestinal Alkaline Phosphatase (TOYOBO) and Exonuclease I (Takara), and directly sequenced with an ABI 3130xl Genetic Analyser (Applied Biosystems and Hitachi, Ltd.) using a BigDye Terminator v3.1 Cycle Sequencing Kit (Applied Biosystems) according to the manufacturer's instructions.

**Phylogenetic analysis for figs**. A final SNP alignment dataset containing 2198 variable nucleotide sites of MIG-seq data obtained from the 60 fig samples was used for phylogenetic analysis. ML analysis was carried out using RAxML v8.2.9[54] with a GTR + Γ model, and bootstrapping was performed with 1000 replicates. The pgsumtree v2.0 (https://github.com/astanabe/Phylogears) was used to output the tree (Newick file). BI analysis was performed using MrBayes v3.2.7a[55] under the GTR + I + Γ model with two separate runs each of four chains for 1,000,000 generations. The trees were sampled every 1000 generations. Convergence of runs was assessed with Tracer v1.7.1[56] and the first 25% of trees were discarded as burn-in, by default.

The plastid DNA and ITS sequences were assembled and manually edited using the GAP4 programme included in the STADEN v2.0.0b9 package[57]. The obtained sequences were aligned using MAFFT v7.310 with L-INS-i model. The alignment (3753 bp) of plastid DNA sequences was used to construct a statistical-parsimony network using TCS v.1.21[59]. The ITS sequences alignment (671 bp) was used for phylogenetic analyses with ML and BI methods. ML analysis was performed on MEGA X[60] under the T92 model (the best-fit model selected by MEGA X) with 1000 bootstrap replicates and complete deletion of gaps. BI analyses were conducted as described above using MrBayes[55] under the GTR + I + Γ model.

**Phylogenetic analysis of pollinators**. As described above, all sequences of the 28 S rDNA, ITS1, and *COI-COII* regions were assembled and manually corrected using the GAP4[57], followed by alignment using MAFFT with L-INS-i[58]. The sequence alignments were then subjected to phylogenetic analysis with ML and BI methods. For ML analysis, the model selection for each dataset was performed on MEGA X[60]. The obtained best-fit models were TN93 + I + Γ for *COI-COII*, TN93 + Γ for 28 S rDNA and HKY for ITS1. ML trees were constructed in MEGA X with 1000 bootstrap replicates and complete deletion of gaps. BI analyses were conducted using MrBayes[55] under the GTR + I + Γ model. To analyze the sequence differences in the *COI-COII* dataset, an NJ tree was also constructed using uncorrected pairwise *p*-distances. Outgroups in three separate genera were used for the analysis of the 28 S rDNA sequence data, and two species, *Eupristina verticillata* and *Platyscapa* sp., which are much more closely related to *Blastophaga*, were used for the phylogenetic analysis of *COI-COII* sequences[16,28].

**Analysis of pollinator divergence times**. A dated coalescent-based species tree was inferred using StarBEAST 2, which is implemented in BEAST version 2.6.4[61] using representative *COI-COII* and 28 S rDNA sequences from pollinator wasps representing each clade in Fig. 3. In addition, 11 species of fig-pollinating wasps in 7 genera were used as outgroups (Supplementary Fig. 10). Substitution models for each gene were inferred during the analysis using the package bModelTest in BEAST 2 to allow for site model uncertainty. Two separate and unlinked clock and tree models corresponding to the *COI-COII* and 28 S rDNA sequences were set. Each model employed a strict clock model, a birth-death tree prior, and analytical population size integration. Following Cruaud et al.[28], the following three calibration points were used to estimate the divergence time of pollinator wasps: split between *Blastophaga* and *Wiebesia* (~40.5 Ma [54.7 − 27.6]), split between *Platyscapa* and *Eupristina* + *Pegoscapus* (~43.5 Ma [55.5 − 32.7]), and the crown of Agaonidae (~75.1 Ma [94.9 − 56.2]) (see Supplementary Fig. 10). Each calibration point was assigned a lognormal prior distribution covering 95% probability range. The MCMC chains were run twice for 40 million generations, storing every 5000$^{th}$ tree, which generated 8000 trees for each run. The tree files of two runs were then combined with LogCombiner (BEAST 2 package). Convergence of runs was assessed with Tracer[56] and trees were generated with the burn-in removed (10%) using TreeAnnotator v2.1.2 (BEAST 2 package). The resulting species tree was visualized with FigTree v1.4.4 (https://github.com/rambaut/figtree).

**Population genetic analyses of figs**. Nine SSR loci were used: Frac241, Frub061, Frub391, Frub416, Frac222[62], FinsM5, FinsQ5[63], FM1-27[64], and FereACOI (Supplementary Table 3). We used the same experimental procedures and data analyses outlined in our previous study[24] with some modifications. Briefly, PCR was performed using a Multiplex PCR plus kit (Qiagen) in 10 µL reaction mixtures containing 1 µL of template DNA. The temperature conditions used for the PCR are shown in Supplementary Table 4. PCR products were analyzed on an ABI 3130 Genetic Analyser with Gene Scan 500 LIZ Size Standard and genotyped with GeneMapper Software, version 4.0. The SSR data were finally obtained from a total of 447 individuals representing all populations of the three fig species, *F. erecta*, *F. formosana*, and *F. vaccinioides* (Table 1 and Supplementary Data 1). GenAlEx v6.502[65] was used to construct SSR datasets for genetic analyses on STRUCTURE v2.3.4[66,67] and Arlequin v3.5[68]. The STRUCTURE programme was used to infer the genetic population structure of the figs and their pollinators and to generate trees with genetic distances shown for the structure clusters[69,70]. The programme

was run with the length of the burn-in period set to 100,000 and the number of MCMC replicates after burn-in set to 100,000. The admixture model for a given $K$ (number of clusters) was simulated using 20 iterations. The number of potential clusters was assessed with $K = 1$–14 for the SSR data. The most likely $K$ of the dataset was obtained by estimating $\Delta K$ value[71] using STRUCTURE_HARVESTER v0.6.94[72]. CLUMPP[73] was used to align and visualize the results of multiple runs. Arlequin[68] was used for population genetic analysis with default settings. The pairwise $F_{ST}$ values among the populations were estimated using the distance method (pairwise differences) and the $F_{ST}$ $P$ values were based on 110 permutations (default setting). In the Arlequin analysis, populations (HK of *F. erecta* and LY of all fig species) containing fewer than 20 individuals were removed from the SSR dataset to avoid obtaining biased results.

**Population genetic analyses of pollinators.** Fifteen SSR loci were used for pollinator wasps: Ega01, Ega03, Ega06, Ega07, Ega08, Ega09, Ega10, Ega11, Hga04, Hga09, Hga10, Eca15, Eca22, Hca07, and Hca19[74] (Supplementary Table 3). PCR primers and conditions are shown in Supplementary Tables 3 and 4. The SSR data were finally obtained from a total of 263 individuals representing all populations of the three pollinator wasp species, *B. nipponica*, "*B. taiwanensis*", *B. yeni*, and *B.* sp. (Table 1 and Supplementary Data 1). The LY and IR populations were removed from the SSR dataset to avoid obtaining biased results because the number of individuals in the populations was less than 20. All procedures for experiments and data analyses were carried out as described in the above section, "Population genetic analyses of figs". The number of potential clusters was assessed with $K = 1$–10 for the analysis of population genetic structure on STRUCTURE v2.3.4[66,67], and the Evanno's $\Delta K$ ($K = 5$) was estimated by STRUCTURE_HARVESTER[72]. Arlequin[68] was used to analyze pairwise $F_{ST}$ values between populations.

**Species delimitation of pollinators.** A PTP model[75] was used to delimit species based on the phylogenetic tree inferred using the *COI-COII* sequences (Supplementary Fig. 5). The NEXUS format of the tree file inferred by MrBayes was converted to Newick format using SeaView ver. 4.7[76], before being applied to the PTP model for species assignment. PTP analyses were performed on the bPTP webserver (a Bayesian implementation of the PTP model for species delimitation, https://species.h-its.org/ptp/) with default settings[75]. The delimitation results were inferred by ML and the highest Bayesian supported solution.

**Cophylogenetic analysis.** eMPRess GUI Version 1.0[77], a software programme for phylogenetic tree reconciliation under the duplication-transfer-loss (DTL) model, was used to determine whether the pairs of figs and their pollinator wasps coevolved with each other by comparing their phylogenetic relationships. The input tree topologies (Supplementary Fig. 12a) of figs and pollinator wasps were based on the results of phylogenetic analyses (Figs. 2 and 3) and population genetic analyses (Fig. 4 and Supplementary Tables 1 and 2). Because eMPRess does not allow multiple host species to be associated with one pollinator species, single host-pollinator associations were used in the analysis, even though the *B. silvestriana* wasps associated with the five fig species are genetically indistinguishable (Fig. 3). The analysis was conducted using the following eMPRess parameters: duplication cost = 1, transfer cost = 2, and loss cost = 1.

**Reporting summary.** Further information on research design is available in the Nature Research Reporting Summary linked to this article.

## Data availability

The sequence data were deposited in the DDBJ/EMBL/GenBank under the following accession numbers: DRR315461-DRR315520 for MIG-seq, LC626904-LC627035 and LC647291-LC647338 for plastid DNA, LC626702-LC626722 for ITS, LC647577-LC647632 for ITS1, LC627753-LC627757 and LC647276-LC647290 for 28 S rRNA, LC627616-LC627750 and LC647633-LC647648 for *COI-COII*. The DNA sequence alignments, the calibrated phylogenetic tree, and SSR data are available at https://doi.org/10.5061/dryad.3n5tb2rj9.

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

## Acknowledgements

We thank Yong Chen, Xiao-Yong Chen, Yu-Fen Cheng, Guan-Ling Sun, Hiu-Wa Lee, Yi-Siun Liu, Yen-Chen Chang, Sin-Hong Pan, Yu-Jyun Yi, Koichi Arimoto, Akane Hoshino and Huizi Wu for assistance with sample collection. We are grateful to Yoshihiro Handa (Bioengineering Lab. Co., Ltd., Kanagawa, Japan) for assistance with the MIG-seq analysis.

## Author contributions

Z.-H.S. designed the study, performed the phylogenetic and population genetic analyses, and wrote the paper with input from the other authors. A.S. generated the sequence and SSR data. J.K. conducted the divergence time estimates and wrote the method. Z.-H.S., A.S., J.K., P.-A.C., H.-Y.T., H.-Q.L., and H.Y. contributed to sample collection and approved the final manuscript.

## Competing interests

The authors declare no competing interests.
