## [Peer Review File · Communications Biology]

Reviewers' comments:

Reviewer #1 (Remarks to the Author):

Totally, it is an interesting story. In present study, Su et al. tested mutualistic relationship of a fig species complex (*Ficus erecta* complex) and their wasp pollinators. They examined the phylogenetic relationships and population genetic structure using MIG-seq data, microsatellite data, chloroplast DNA, mitochondrial DNA, and nuclear 28S rDNA data. By this, they showed the complex mutualistic pattern and mechanisms of diversification and speciation of Blastophaga wasps.

Some comments and suggestions are as following:

Line 20: please add "Agaonidae, Hymenoptera" after "wasps"

Lines 52-54: do you mean tribe Castilleae has no "obligate mutualistic relationship" with other insects? it should be pointed out clearly.

Line 67: "8,16,17-20" should be "8,16-20"

Line 101: as you noted, *Ficus erecta* complex comprises approximately 17 closely related species, why and HOW did you select six of them in the present study? Are they the most closely-related ones in this complex?

Lines 133-137, Fig S1: the monophyly of *F. erecta*, *F. formosana* and *F. vaccinioides* was not supported based on chloroplast DNA data, does that mean there is some introgression, if yes, a discussion could be provided; it is better to label clade 1, 2, 3 on Fig S1.

Lines 151-156: Clade 4 contained individuals of *B. taiwanensis* and *B. nipponica*, and clade 5 only contained individuals of *B. nipponica*, so the monophyly of these two species was not supported. Does that mean there is introgression or incomplete lineage sorting between *B. taiwanensis* and *B. nipponica*? Based on both mt DNA and SSR data (e.g., Fig 3, Fig S2, Fig 4), the wasp *B. taiwanensis* seems not a "species", is there more evidence other than host shift? As your address, five fig species share a single pollinator species *B. silvestriana*, why not *F. erecta* and *F. formosana* can share "*B. nipponica*"(including "*B. taiwanensis*")?

Lines 157-162: in fact, the nuclear 28S gave little phylogenetic information, I think it is hard to say "consistent with" the results getting from mt DNA data.

line 247: the reference is 35, not 29.

Line 236: should *F. vaccinioides* be deleted here?

Reviewer #2 (Remarks to the Author):

Please see attached review comments.

Reviewer #3 (Remarks to the Author):

In this contribution, the authors investigate the diversification history of a closely related group of figs and their pollination wasps. While traditionally thought to be extreme examples of co-diversification, interactions between figs and their pollinating wasps are now thought to be more complex. Recent studies have shown the existence of multiple co-occurring pollinators for a given fig and also of pollinator sharing between figs. Considering that dispersal to islands presents opportunities for the disruption of strict fig/wasp interactions, the authors describe the recent phylogenetic history of figs and wasps found in the Ryukyu Islands, Taiwan and their mainland counterparts. They find extensive pollinator sharing on the mainland, and also evidence for multiple host shifts between wasps. While, as the authors point out, it is not a novelty that such processes exist, to my knowledge they have never been demonstrated with this geographical and phylogenetic extent, and particularly never making use of islands as natural experiments.

I commend the authors for the extensive geographical sampling and also the large sampling of figs, wasps and multiple kinds of genetic data. However, I believe that all of this evidence was not combined in the most clear way, resulting in a confusing manuscript in which the main points seem to be lost at times. Moreover, there are methodological problems with phylogenetic analyses

and the discussion seems to rely more on arbitrary interpretations of these phylogenies than on reproducible analyses (e. g. biogeography and co-speciation). For those reasons, I believe that the manuscript requires a significant revision prior to being published.

Major comments:

1) Usage of different kinds of data

The authors collected multiple kinds of molecular data and used each kind for different analyses, but in general there are no good justifications on why certain kinds of data were used for some analyses.

For example, the SNP dataset collected with MIGseq could have been used not only for fig phylogenetics, but also for population structure using nuclear markers and for investigating the origin of the *F. erecta* complex distributed in Taiwan-Ryukyu Islands (line 130). For example, the tree based on MIGseq in Fig. 2 contradicts the plastid tree in Fig. S1a (assuming correct rooting, see below), both are compatible with the unrooted tree from SSR distances in Fig. 4a. Why is the plastid 3 better for this question? Interpretation of Fig. 2 (which is the chosen tree in Fig. 5) suggests that each of the species independently dispersed to the islands or from the islands to mainland, while the plastid 3 indicates that *F. erecta* is paraphyletic and that the other species are closely related to *F. erecta* from GD, which is the interpretation adopted in the paragraph starting in line 130. Maybe this discordance shows something about the differential dispersal of pollen and seeds?

This is one example, but more generally I would expect a better justification of why using certain kinds of data for certain questions. In the example above, it actually seems more appropriate to consider data from all sources.

2) Phylogenetic tree rooting

Clades and the relationships between them are used as important evidence throughout the manuscript. However, with the exception of the calibrated tree in Fig S5, none of these trees is rooted with an outgroup and therefore they cannot provide information on direction of events. The recent example of SARS-Cov-2 phylogeny and its difficult interpretation highlights the limitations of making inferences on directionality of dispersal and host shift events without a proper root (see Pipes et al, 2020, <https://doi.org/10.1093/molbev/msaa316>). Here in this paper we see similar difficulties. For example, in line 193 the authors interpret Fig. 4a as showing that the *F. erecta* population in GD and HK is distantly related to the others. This interpretation is valid under a strict molecular clock (implying mid-point rooting), but not if the tree is rooted in one of the short branches separating the 3 populations of *F. erecta*. In general, I would advise to justify rooting and, when data is available, including outgroups. In Fig. 2 this is hard, but could be done if whole genome data is available for a closely related fig species, for example. In the trees based on more traditional markers (chloroplast genes, COI, etc), there is most likely data available for outgroups in Genbank. By properly rooting these trees, it is possible that the conclusions found in the paper based on an arbitrary root would be different

3) Calibrated phylogenetic tree

The authors make use of the divergence times estimated to discuss the evolution of geographical range of fig and wasp species. In the absence of any closely related fossil, I doubt that it is possible to obtain estimates that are precise enough for this, but in any case I do not believe this is a strictly part of the paper.

However, if the authors do opt to continue relying on a calibrated phylogenetic tree, several problems with the current tree must be corrected.

The calibration point used is not well-justified. The crown age of the 4 species of *Blastophaga* included in Cruaud et al. is 15 million years, not 10 million years as used here. Therefore, one assumes that the node used for calibration is the split between *B. nipponica* and two undescribed species of *Blastophaga*, obtained from *Ficus stenophylla* and *F. chapaensis*. Since neither of these are included, it is not clear why this node in Cruaud et al corresponds to the crown group of the species sampled here. For example, If all species here are actually more closely related to *B. nipponica* than to any other species sampled in Cruaud et al, then this would be overestimating the age of the group. Alternatively, if there are species included here that do not belong to the clade in Cruaud et al, this would be underestimated. One indication that something is wrong is that the split between *Wiebesia* and *Blastophaga* was inferred to ~20 million years (Fig S5), half of the age of the equivalent node in Cruaud et al. In fact, this makes me wonder why the splits between

genera were not used as calibration points, since their correspondence between analyses seems much more straightforward.

Another problem with the analysis is the topological prior used (Yule prior). This assumes no extinction, complete sampling, and that tips correspond to species. Because multiple samples per species are used, the latter is violated. Complete sampling is also certainly violated. To accommodate the multiple samples per species, I would suggest clustering sequences in species corresponding to the clades identified and using clades as tips in an analysis using the multispecies coalescent model in STARBEAST2 instead of a concatenated analysis with each sample as tip. Together with a more appropriate prior (for example, birth-death) and using the splits between genera as calibration points, this should produce more reliable estimates. Finally, there is not need to assume a GTR+GAMMA model in STARTBEAST2 (RAxML only has that option), and the substitution model can be inferred together with the tree with the BEAST2 package bmodeltest (see <https://github.com/BEAST2-Dev/bModelTest/wiki>)

4) Species delimitation

Especially for wasps, the authors split groups in the phylogeny arbitrarily as "clades". It seems to be this is a missed opportunity for a more reproducible analysis or criterion that clearly identifies what are these clades. In line 246, for example, the authors discuss that the % difference between samples is close that what is typically considered as different species. Are they different species of fig wasps? I would suggest the usage of tree-based species delimitation methods instead of arbitrary definition of clades. Some examples of possible methods are:

STACEY (BEAST2 package: <https://www.beast2.org/2015/06/02/species-delimitation-with-beast.html>)

Jones G. Algorithmic improvements to species delimitation and phylogeny estimation under the multispecies coalescent. *J Math Biol.* 2017 Jan;74(1-2):447-467. doi: 10.1007/s00285-016-1034-0. Epub 2016 Jun 10. PMID: 27287395.

GMYC (for example, available in <https://species.h-its.org/gmyc/>)

Tomochika Fujisawa, Timothy G. Barraclough, Delimiting Species Using Single-Locus Data and the Generalized Mixed Yule Coalescent Approach: A Revised Method and Evaluation on Simulated Data Sets, *Systematic Biology*, Volume 62, Issue 5, September 2013, Pages 707–724, <https://doi.org/10.1093/sysbio/syt033>

With this delimitation, it would be easier to discuss the patterns observed, and the species limits could also be used for a multispecies coalescent model for the calibrated tree as I suggest above.

5) Biogeography and cospeciation

Just like with species delimitation, the discussion of biogeographical implications of the phylogenetic trees presented relies on arbitrary interpretation rather than reproducible methods. I suggest that the authors used their ultrametric phylogenetic trees in a program such as BioGeoBEARS (see <https://github.com/nmatzke/BioGeoBEARS>) or RevBayes (see <https://github.com/nmatzke/BioGeoBEARS>) to infer the biogeographical history of figs and wasps. Even if no reliable calibration is possible and the ultrametric tree has branch lengths in units of substitution per site, this would be better than arbitrary interpretation.

For example, I believe it would show migration of wasps not only between mainland and islands (the only ones in Fig. 6), but also between islands and the mainland, since *B. yeni* corresponds to a paraphyletic group of two species.

The same can be said about inferences on host shifts. See RevBayes (https://revbayes.github.io/tutorials/host_rep/host_rep.html) and eMPress (<https://sites.google.com/g.hmc.edu/empress/home>) for cophylogeny reconciliation.

6) Introgression

In lines 270-274, the authors discuss whether or not sharing of fig pollinators could lead to introgression between fig species. The incongruence between trees from plastid genes and nuclear reduced-representation data may indicate that there is some degree of introgression but this is not evaluated in this paper. If MIG-seq is similar to other reduced representation libraries, it should provide enough data to address this question. For example, a recent paper also found evidence for host switches in fig wasps and in that case they led to hybridization between fig species. This

paper is not currently discussed in this manuscript, but it seems to be relevant: Wang, G., Zhang, X., Herre, E.A. et al. Genomic evidence of prevalent hybridization throughout the evolutionary history of the fig-wasp pollination mutualism. *Nat Commun* 12, 718 (2021). <https://doi.org/10.1038/s41467-021-20957-3>.

I suggest the authors to consider whether a similar kind of analysis could be performed here.

7) General formatting

The analysis of pollen in wasp bodies mentioned in the discussion in line 269 is not described in methods or results.

I believe the journal instructs authors to finalize the introduction with a summary of major results.

Minor comments:

line 51

"specious" - probably meant "speciose"

line 115 since there were so many different types of sequencing methods used, I suggest that these numbers are presented in a supplementary table and that the method that corresponds to these reads is stated explicitly (MIG-seq). More than raw reads, I would be interested in knowing the number of sites (which is reported) and the amount of missing data in the matrix (not reported currently). Is missing data biased with respect to phylogeny, as in other reduced representation methods such as RADseq?

line 181 - why not include it as supplement?

line 269 - please add details in methods and results

line 334 - see comments on divergence dating. Regardless, the confidence interval is large enough to encompass this event

line 485 - SNs - probably meant SNP?

Fig 4 - This figure is really hard to interpret. Colors are confusing since the same colors do not correspond to the same populations. I would suggest to use similar colors for similar locations to the extent possible. This would make comparison of Fig 4a and 4d easier, for example. I would also suggest a single value of K in Fig. 4b. Maybe the other values could be shown in a supplementary figure.

Fig. 5 - This is a nice conceptual image, but the migration and host shift events are based on arbitrary interpretation rather than reproducible analyses, see comments above. I suggest redoing this figure after results of biogeographical and cophylogeny analyses

Fig. 6 - same as Fig. 5

Fig. S1 - Please use the same colors for species in panels A and B

Fig. S4 - Please indicate the ingroup (and mention in the appropriate section in the methods that outgroups were used and how they were chosen)

Fig. S5 - Please add a time axis in bottom, which makes visualization easier

Tables S5-S6 - maybe 3 decimal places would suffice and make more readable?

Responses to the reviewers' comments

Reviewer #1 (Remarks to the Author):

Totally, it is an interesting story. In present study, Su et al. tested mutualistic relationship of a fig species complex (*Ficus erecta* complex) and their wasp pollinators. They examined the phylogenetic relationships and population genetic structure using MIG-seq data, microsatellite data, chloroplast DNA, mitochondrial DNA, and nuclear 28S rDNA data. By this, they showed the complex mutualistic pattern and mechanisms of diversification and speciation of *Blastophaga* wasps.

Reply: Thank you for your positive review of our manuscript.

Some comments and suggestions are as following:

Line 20: please add “Agaonidae, Hymenoptera” after “wasps”

Reply: We have added. **Line 20.**

Lines 52-54: do you mean tribe Castilleae has no “obligate mutualistic relationship” with other insects? it should be pointed out clearly.

Reply: Thank you for your kind comments.

The text regarding the references to the Castilleae has been removed, and added some other sentences in the revised version. **Lines 53-55.**

Line 67: “8,16,17-20” should be “8,16-20”

Reply: We have changed. The numbers have been changed to “8,28-32” in the revised version. **Line 70.**

Line 101: as you noted, *Ficus erecta* complex comprises approximately 17 closely related species, why and HOW did you select six of them in the present study? Are they the most closely-related ones in this complex?

Reply: Thank you for your kind comments.

The six species were considered to be suitable for this study, because 1) they are distributed in Taiwan-Ryukyu islands and/or the adjacent continent (mainland China) and 2) they are most closely related with each other in the *F. erecta* complex (Ref. 37: Lu et al., 2017). We have added such explanation to the section, *Sampling* in Methods. **Lines 560-563.**

Lines 133-137, Fig S1: the monophyly of *F. erecta*, *F. formosana* and *F. vaccinioides* was not supported based on chloroplast DNA data, does that mean there is some introgression, if yes, a discussion could be provided; it is better to label clade 1, 2, 3 on Fig S1.

Reply: Thank you for your kind remarks.

The not-supported-monophyly of the three species is probably due to the too small sequence differences within these closely related species. You may see from chloroplast DNA network, in which only one or a few bases of sequence differences were found among these species (Cluster 1 in Fig. S2). However, as you pointed out, some introgression may have occurred, as some *F. erecta* individuals had same haplotype sequence with that of *F. formosana* (h9 in Fig. S2). Therefore, we have added discussion for this point. **Lines 521-526.**

Because of the small sequence differences and low support values for the phylogenetic tree. Therefore, we removed the tree and left only the network diagram in the revised version, because network diagram gave the number of base substitutions between the haplotype sequences.

In the revised manuscript, the number of this figure was changed to Fig. S2, and the three clades are labeled with Cluster 1, Cluster 2, and Cluster 3 on the figure (we used the name as “Cluster” but not “Clade”, this is to avoid confusion with the clades of pollinator wasps). In addition, haplotypes are indicated by the numbers h1-h24, which are also added into Table S1.

Lines 151-156: Clade 4 contained individuals of *B. taiwanensis* and *B. nipponica*, and clade 5 only contained individuals of *B. nipponica*, so the monophyly of these two species was not supported. Does that mean there is introgression or incomplete lineage sorting between *B. taiwanensis* and *B. nipponica*? Based on both mt DNA and SSR data (e.g., Fig 3, Fig S2, Fig 4), the wasp *B. taiwanensis* seems not a “species”, is there more evidence other than host shift? As your address, five fig species share a single pollinator species *B. silvestriana*, why not *F. erecta* and *F. formosana* can share “*B. nipponica*” (including “*B. taiwanensis*”)?

Reply: Thank you for your kind remarks.

B. taiwanensis is described based on morphological characters (Ref. 38: Chen and Chou, 1997), but the morphological differences between *B. taiwanensis* and *B. nipponica* are

small and it is difficult to distinguish these two species. As you pointed out, the species was also not supported by molecular evidence in the present study although a slight genetic difference was found between the two species (Table S3). We agree with you, it is possible that *F. erecta* and *F. formosana* share *B. nipponica* including “*B. taiwanensis*” at present. Therefore, we have added “” for *B. taiwanensis* throughout the paper and revised the discussion. **Lines 202-204; Lines 506-535.**

Lines 157-162: in fact, the nuclear 28S gave little phylogenetic information, I think it is hard to say “consistent with” the results getting from mt DNA data.

Reply: Thank you for your kind comments.

In the revised version, we additionally sequenced ITS1 region of rRNA gene to provide more strong molecular evidence. The text of the results has been revised. **Lines 205-220.** Please also refer to Fig. S6B.

line 247: the reference is 35, not 29.

Reply: Thank you very much. We have revised. **Line 339.**

In the revised manuscript, some reference numbers have been changed.

Line 236: should *F. vaccinioides* be deleted here?

Reply: *F. vaccinioides* should be included here. We changed the “The five *Ficus* species...” to “The six *Ficus* species...”. **Line 327.**

Reviewer #2 (Remarks to the Author):

Overall, I find this a very interesting and detailed study of the evolutionary history of the fig-fig wasp mutualisms in these clades. Using molecular data and population genetics, the authors describe pollinator sharing and host shifts to reconstruct the mutualistic relationships of these clades. This reconstruction provides an insight into the evolutionary trajectories of mutualistic relationships. The strength of the paper is that the authors provide a detailed molecular study of both lineages, with implications for all systems of brood site pollination mutualisms and the study of evolution and speciation as a whole. This paper is current, relevant and broadly interesting. Method descriptions are sufficient to reproduce the work, except for the inclusion of the statistical tests and corrections for multiple tests of F_{st} . Those details were missing. I find the results supported except that I am concerned with all results regarding *B. taiwanensis*. The molecular data and morphological data do not seem to support this as a different species from *B. nipponica* on TW and so all discussions of this species need to be tamed down and clarified with the caveat of what this data can show. It seems that this is all based on just calling them different names based on the host plant they were collected from and finding a very small F_{st} (0.02). There is much higher F_{st} between populations of *B. nipponica* and as mentioned above, we do not have the methods for the statistical test nor any information of whether these tests have been corrected for multiple comparisons. The title and the abstract need to be reworded to remove the highlighting of this *B. taiwanensis* speciation event that the data does not strongly support. The authors already have a strong paper without overstepping in that discussion. The authors should also be very careful about the discussion of species-specificity in the fig-fig wasp system. We know, and they show, that this is often not the case, yet reference that quite often including the abstract. The wording of the introduction is a bit confusing at times as it not clear if the authors are talking about species diversity in plants or pollinators or both, and it seems to suggest that the results will discuss the implication that pollinator duplication and host shift has on plant diversity and this is not the case; the focus is on the pollinators. The introduction can be reworded to clear this up. I would also like clarity in the introduction whether these are monophyletic clades with all species represented.

Reply: Thank you very much for your positive review and kind remarks of our manuscript.

Introduction

When reading the introduction, I expected the paper to describe ‘species diversity generation’ (Line 109) in both lineages, but really it is in the pollinators. The introduction needs to be gone back over to make sure that the reader’s expectations are correct. The discussion of the high level of diversity in the plants is one such set up that confuses the readers expectations.

Reply: Thank you for your comments.

We have significantly revised the Introduction and Discussion with a focus on the pollinators. We hope that the revised Introduction and discussion will be more in line with the contents of this paper and would not confuse the readers expectations.

Line 54-57 – This sentence sounds as though the assumption is that the obligate mutualism is the cause of the higher number of species of Ficus as compared to Castilleae. There are many possible contributors to the difference in species richness, the pollination mechanism is just one.

Reply: Thank you for your kind comments.

The text regarding the references to the Castilleae has been removed, and added some other sentences in the revised version. Lines 53-55.

Line 58-62 – More recent work has shown that they are not one-to-one as you cite in lines 64-67. Starting the paragraph with this sentence suggests that is the thesis of the paragraph and is confusing to the reader that does not know the literature. You can just remove these lines. This may be the history of the research in the groups, but that needs to be made clear if you choose to leave them in. If you leave them in, provide a citation for ‘the expectation that co-speciation has occurred extensively...’

Reply: Thank you for your kind comments and suggestions.

We have revised the sentence. Lines 57-61.

Line 75 – ‘these speciation processes’

Reply: We have revised as follows.

‘this speciation process’ ⇒ ‘these speciation processes’ Line 75.

Line 76-77 – please define speciation mechanisms here for clarity. Eg. ‘...speciation mechanism, sympatric or allopatric,...’

Reply: Thank you for your kind suggestions.

We have revised as follows.

‘...speciation mechanisms...’ ⇒ ‘...speciation mechanism, sympatric or allopatric, ...’
Line 77.

Line 77 – cite consideration of host shifting as a trigger

Reply: We have significantly revised this part. Please refer to the text of **Lines 78-89.**

Line 80 – ‘...observed in this obligate...’

Reply: This sentence has been removed in the revised version. Please refer to the text of **Lines 78-89.**

Methods

It is not clear by reading which analyses were done on figs and which were done on wasps. I suggest either separating and having two subsections on fig molecular data and wasp molecular data or keeping the current structure and clearly stating in each analysis what was done for figs and what was done for wasps.

Reply: Thank you for your kind suggestions.

According to your suggestions, we have revised the text with keeping the current structure and clearly stating what was done for figs and what was done for wasps. Please refer to the text of Methods.

Line 416 – Is 1 syconia enough to capture all the pollinator species present on the tree? How does your sampling match the know species distributions? What is the total sampling from each population?

Reply: Thank you for your kind comments.

The pollinator samples used in this study were collected with the following points. If possible, 1) collecting one to five syconia per fig tree, and 2) collecting syconia from more than 20 different fig trees for each geographic population. To avoid using pollinator wasps from close relatives, we used wasp samples from different syconia that were collected from different fig trees where possible.

The sample numbers of figs and pollinator wasps from each population of each species are shown in Table 1, and their details are shown in Table S1.

The fig species used in this study are mainly distributed in southern China and/or Taiwan-Ryukyu islands. Our sampling for this study fully covered the distribution

ranges of the Taiwan-Ryukyu islands and included two geographical areas of the continental distribution, the southern China (Guangdong and Hong Kong) and southeastern China (Fujian). Although our sampling could not cover the entire distribution ranges of mainland China, we do not think it would affect our main conclusions described in the Abstract.

The explanation mentioned above has been added in the revised manuscript for the description of sampling. **Lines 572-584 and Lines 595-600.**

Line 446 – In this section did you do MIG-seq on wasps? The introduction to the Methods section suggested you did.

Reply: Thank you for your kind comments.

We did not do MIG-seq on wasps. We have revised the text to make clear this point. **Line 617.**

Line 448 – Did you only do 60 fig samples total or 60 for each run? Ah, I see down in the phylogenetic section that you only did 60. Please clarify how many are used for which analyses in the introduction to the methods section where you list the number of samples, or state it clearly at the beginning of each molecular analysis section. I am confused where the samples are going.

Reply: Thank you for your kind comments.

We did 60 fig samples in total. We have revised the text throughout the manuscript to make clear this point. **Lines 619-623.** Please also refer to Table 1.

Line 472-476 – Please separate into different sentences the genes amplified for the plants and those for the wasps. It would help to clarify for the reader. How many samples of each were used?

Reply: Thank you for your kind suggestions.

We have revised the text in Methods to make clear this point by separating the descriptions for figs and fig wasps into two sections, “*PCR of plastid DNA and ITS sequences of figs*” and “*PCR of 28S rDNA, ITS1, and COI-COII sequences in pollinators*” in Methods. **Line 647 and Line 667.**

Line 485 – Why did you only do ML trees? Some of your support values are low, you should also run MCMC trees.

Reply: Thank you for your kind comments.

We have added MCMC analysis for all sequence datasets.

Please refer to the Methods sections, *Phylogenetic analysis for figs* (Line 683) and *Phylogenetic analysis of pollinators* (Line 702), the relevant results, and also the phylogenetic trees (Figs. S1, S3, S4B, S6A, and S6B).

Line 503 – You did not use all of the individuals for this MCMC analysis? Why?

Reply: Thank you for your kind comments.

You may see in Fig. 3 that the sequence differences within each clade are very small, and some are identical sequences. In this analysis, we focused on estimating the divergence times between clades, but not within clade, so, to shorten analysis time on computer, we selected respective samples from each clade for this analysis. We don't think that the analysis would give essentially different results on the estimation of divergence times between clades with the selected samples or with all the samples. In the revised version, re-analysis was performed by adding outgroup species and calibration points. Please refer to Supplementary Fig. S8.

Line 512-516 – move the calibration point sentence up to line 508, before LogCombiner.

Line 540 – ‘of figs and fig wasps’

Reply: Thank you for your kind suggestions.

The text for this section has been largely revised. The sentence about ‘calibration point’ was moved to the place you suggested. Please refer to the section, “*Analysis of pollinator divergence times*” in Methods. Lines 716-736.

For Line 540 – ‘of figs and fig wasps’, we have revised.

The Method text including this sentence has been largely revised. Please refer to the Methods sections, “*Population genetic analyses of figs*” and “*Population genetic analyses of pollinators*”. Lines 738 and Line 765.

Line 538-543 – what statistical test did you perform to test for significance of these F_{ST} values? Did you do any multiple test corrections?

Reply: Thank you for your kind comments.

The F_{ST} P values were based on the number of permutations (default setting: 110) on the Arlequin v3.5, and this sentence has been added to the text in Methods. Lines 759-761.

Results

Line 133 – This is not throughout their distributions as you do not include any samples north of Ishigaki Island, leaving out five populations. Why didn't you include samples from these populations?

Reply: Thank you for your kind comments.

The samples north of Ishigaki Island are geographically distant from the continental populations and was not considered necessary for the purpose of this analysis.

According to your suggestions, in the revised version, we reanalyzed the network by including these samples. Please refer to Supplementary Fig. S2 and the revised results.

Lines 148-165.

Line 233 and on – you have not introduced what the abbreviation y-TW means or subsequent n-IS n-YN. I'm guessing it is the species name, but please make this clear.

Reply: Thank you for your kind comments.

As you guessed, the abbreviations y-, n-, and t- mean the species name, which are explained in Table S3. In the revised version, we removed y-, n-, and t-, and left only the population names in the result text. We think that readers can understand the contents by referring to Table S3. Lines 302-310.

Discussion

Line 270-274 – or pollen incompatibility

Reply: We have added this possibility. Lines 369-370.

Line 291-292 – ‘... and so this is the first record of *B. nipponica* collected from’

Reply: We have revised. Lines 394-395.

Line 296 – fig 6d – there are a few subsequent moments where you mention fig 6, but not the panel in that figure, please do so throughout.

Reply: Thank you for your kind comment. We have done that.

Line 297-301 – While your suggestion makes sense, it is also possible that sampling has missed it. Failure to detect does not equal absence and should be noted.

Reply: Thank you for your kind comment.

We have added the sentence “failure to collection does not equal absence” to the text in the revised version. Lines 399-402.

Line 304 – change ‘considered’ to ‘seems’

Reply: We have changed. Line 408.

Line 317 – change ‘are’ to ‘can be’

Reply: We have changed. Line 430.

Line 320-327 – given your current study and the existing research, we know that the relationships are not species specific.

Reply: We have made some modifications to the text around here. Lines 432-439.

Line 359-360 – elaborate and reference figures.

Reply: We have made some modifications to the text around here, and added reference figures. Lines 487-492.

Line 379 – remove ‘by chance’

Reply: We have removed. Line 510.

Line 380 – remove ‘adaptive evolution’

Reply: We have removed. Lines 511-512.

Line 374-389 – your molecular results suggest that *B. taiwanensis* and *B. nipponica* are the same species and they were difficult to tell apart by morphology. Why are they considered different species at all? I think that this portion of the results need to be tamed down. The results would clearly suggest that speciation has not been completed.

Reply: Thank you for your kind comments.

Indeed, our results do not support that *B. taiwanensis* is a distinct species from *B. nipponica*. However, it is certain that there is a host shift of *B. nipponica* from *F. erecta* to *F. formosara*. We have made major revisions to this portion of the discussion by separately discussing the outcomes of the two host shifts, i.e. the pollinator of *F. vaccinioides* to *F. erecta*, and the pollinator of *F. erecta* to *F. formosara*. We hope that the revised discussion could gain your understanding. Please refer to the text of the section, *Host-specificity reestablishment leading to host-shift speciation* in Discussion. Lines 501-535.

Line 388-389 – Species differentiation is more than host selection, there are many

biological and behavioral things that may contribute to reduced interbreeding when encountering each other. Host selection would only tell you if they encounter each other and speciation is not required for host plant specialization to be occurring.

Reply: Thank you for your kind comments.

We agree with you that host selection testing cannot determine whether speciation has been completed or is in progress. We have revised the discussion. Lines 534-535.

Line 399 – ‘...the establishment of some species-specific...’

Reply: The sentence has been revised. Line 552-553.

Figures

Fig 3. You did a MCMC run with Beast, why are there only ML bootstrap values here? You mention in methods that each caption has the model used, but that is not listed in this figure caption. Caption should also mention that the topology matches the 28s data and reference the two sup figures.

How are Fig 3 and Supplementary Fig S2 different? It seems that Fig S2 is just repeated with more information in Fig 3.

Reply: Thank you for your kind comments.

In the revised version, we added a MCMC analysis with MrBayes (Bayesian Inference, BI). The results are shown in Figs. S4B. The results gave the same tree topology among the clades with that obtained by ML and NJ analyses (Fig. 3 and Fig. S4A).

Since the 28S data had no phylogenetic information among the Taiwan-Ryukyu lineages of *Blastophaga* species, in this revised version, we added ITS1 sequence data and performed the phylogenetic analyses. The tree topology matches that based on the COI-COII data, except for the grouping of two clades of pollinators (*B. yeni* and *B. sp.*) associated with *F. vaccinioides* (Fig. S6B). These results and the relative discussion were added in the text of the revised version. Lines 205-220 and Lines 340-344.

The models for phylogenetic analyses are described in the text of Methods. Lines 683-714.

We have mentioned in caption of Fig. 3 that the topology matches the 28S and ITS1 data and reference the two supplementary figures.

Fig 4. I like this figure, very clearly laid out. I don't think that K=5 or K=7 are necessary here, they could go in the supplementary materials, but it doesn't take away from the figure for me if you want to leave it in. However, I would prefer that the colors of the clusters in a) match those of K=6 in b). You should also label the population section between b) and c) for readers that are skimming the paper.

Reply: Thank you for your positive comments. We have revised this figure according to your suggestions. Please refer to Fig. 4 and Fig. S9.

Fig 5. This figure has a lot of information and you did a good job getting it all on there. This summarizes the paper nicely.

Reply: Thank you for your positive comments.

Fig 6. State in caption where you are getting that date from in a). Caption should walk reader through e).

Reply: We have revised the caption of this figure according to your suggestions. Please refer to Fig. 6.

Fig S5. Please put time scale along the bottom of the figure, it will make it easier to read.

Reply: We have reanalyzed the divergence times of the phylogenetic lineages of pollinator wasps by adding more outgroups and calibration points. We have also put the time scale along the bottom of the figure. Please refer to Fig. S8.

Figure S6. What do the colors mean? What do the sample names mean? This figure is not clear.

Reply: We are sorry for the difficult in understanding this figure. We have revised this figure and its caption. We hope the revised figure will be easier to understand. Please refer to the text of Methods (Lines 652-660), Results section, *Analysis of ITS sequences of fig pollen carried by foundresses* (Lines 167-183), and Fig. S3.

Sup Table 1 - Header 'pollinating fig wasps'

Reply: We have corrected. In the revised version, we changed this Sup Table S1 to Table 1.

Reviewer #3 (Remarks to the Author):

In this contribution, the authors investigate the diversification history of a closely related group of figs and their pollination wasps. While traditionally thought to be extreme examples of co-diversification, interactions between figs and their pollinating wasps are now thought to be more complex. Recent studies have shown the existence of multiple co-occurring pollinators for a given fig and also of pollinator sharing between figs. Considering that dispersal to islands presents opportunities for the disruption of strict fig/wasp interactions, the authors describe the recent phylogenetic history of figs and wasps found in the Ryukyu Islands, Taiwan and their mainland counterparts. They find extensive pollinator sharing on the mainland, and also evidence for multiple host shifts between wasps. While, as the authors point out, it is not a novelty that such processes exist, to my knowledge they have never been demonstrated with this geographical and phylogenetic extent, and particularly never making use of islands as natural experiments.

I commend the authors for the extensive geographical sampling and also the large sampling of figs, wasps and multiple kinds of genetic data. However, I believe that all of this evidence was not combined in the most clear way, resulting in a confusing manuscript in which the main points seem to be lost at times. Moreover, there are methodological problems with phylogenetic analyses and the discussion seems to rely more on arbitrary interpretations of these phylogenies than on reproducible analyses (e. g. biogeography and co-speciation). For those reasons, I believe that the manuscript requires a significant revision prior to being published.

Reply: Thank you for your kind remarks.

Major comments:

1) Usage of different kinds of data

The authors collected multiple kinds of molecular data and used each kind for different analyses, but in general there are no good justifications on why certain kinds of data were used for some analyses.

For example, the SNP dataset collected with MIGseq could have been used not only for fig phylogenetics, but also for population structure using nuclear markers and for investigating the origin of the *F. erecta* complex distributed in Taiwan-Ryukyu Islands

(line 130). For example, the tree based on MIGseq in Fig. 2 contradicts the plastid tree in Fig. S1a (assuming correct rooting, see below), both are compatible with the unrooted tree from SSR distances in Fig. 4a. Why is the plastid 3 better for this question? Interpretation of Fig. 2 (which is the chosen tree in Fig. 5) suggests that each of the species independently dispersed to the islands or from the islands to mainland, while the plastid 3 indicates that *F. erecta* is paraphyletic and that the other species are closely related to *F. erecta* from GD, which is the interpretation adopted in the paragraph starting in line 130. Maybe this discordance shows something about the differential dispersal of pollen and seeds?

This is one example, but more generally I would expect a better justification of why using certain kinds of data for certain questions. In the example above, it actually seems more appropriate to consider data from all sources.

Reply: Thank you for your kind comments.

We have added some explanations to relevant sections of Methods and Results for each kind of analyses. We hope the revised version will be easier to understand.

As you pointed out, MIG-seq data can also be used for population genetic structure analysis, but in this study, we obtained MIG-seq data only from the representative samples from each population of each fig species. These data are insufficient for population genetic structure analysis, so we used MIG-seq data to do phylogenetic analysis and SSR data to do population genetic structure analysis.

For plastid DNA data, we are sorry that the description in the text seemed to be a little misleading. In plants, both seed and pollen dispersal will cause gene flow between geographical populations. We tried to analyze plastid DNA to confirm whether there is any differential dispersal of seeds and pollen. As expected, some inconsistent results were found between nuclear and plastid DNA data, although the plastid DNA data could not give detailed and robust phylogenetic relationships among the species and geographical populations due to small sequence differences. We have revised the description in the text of Results (Lines 148-165) and the discussion in the sections, *Establishment of F. erecta complex and their pollinators in Taiwan-Ryukyu islands* (Lines 468-483), and *Host-specificity reestablishment leading to host-shift speciation* (Lines 521-524).

For the tree topologies of figs and pollinators shown in Fig. 5, the fig tree is based on the results obtained from MIG-seq and SSR data (Figs. 2 and 4a), and the pollinator tree

is based on the results inferred from COI-COII sequences and SSR data (Figs. 3 and 4d). These descriptions have been added to the caption of the figure.

As mentioned above, plastid DNA data could not give robust phylogenetic relationships among the fig species or populations, so we have removed the phylogenetic tree, and left only the network figure in the revised version (Fig. S2).

2) Phylogenetic tree rooting

Clades and the relationships between them are used as important evidence throughout the manuscript. However, with the exception of the calibrated tree in Fig S5, none of these trees is rooted with an outgroup and therefore they cannot provide information on direction of events. The recent example of SARS-Cov-2 phylogeny and its difficult interpretation highlights the limitations of making inferences on directionality of dispersal and host shift events without a proper root (see Pipes et al, 2020, <https://doi.org/10.1093/molbev/msaa316>). Here in this paper we see similar difficulties. For example, in line 193 the authors interpret Fig. 4a as showing that the *F. erecta* population in GD and HK is distantly related to the others. This interpretation is valid under a strict molecular clock (implying mid-point rooting), but not if the tree is rooted in one of the short branches separating the 3 populations of *F. erecta*. In general, I would advise to justify rooting and, when data is available, including outgroups. In Fig. 2 this is hard, but could be done if whole genome data is available for a closely related fig species, for example. In the trees based on more traditional markers (chloroplast genes, COI, etc), there is most likely data available for outgroups in Genbank. By properly rooting these trees, it is possible that the conclusions found in the paper based on an arbitrary root would be different

Reply: Thank you for your kind comments.

For the Fig. S5 in the old version, we have re-analyzed the divergence time of the *Blastophaga* wasps with 11 outgroup species representing 7 genera of fig-pollinating wasps (Fig. S8). For the phylogenetic analyses of figs based on MIG-seq data, two species, *F. nipponica* and *F. thunbergii* were used as outgroups (Fig. 2), and in the pollinator phylogenetic analyses based on COI-COII sequences, *Eupristina verticillate* and *Platyscapa* sp. were used as outgroups (Fig. 3 and Fig. S4).

For the detailed explanations, please refer to the text of Methods, Lines 621-623 in section *MIG-seq experiment for figs*, Lines 667-675 in section *PCR of 28S rDNA, ITS1, and COI-COII sequences in pollinators*, Lines 719-720 in section *Analysis of pollinator divergence times*.

3) Calibrated phylogenetic tree

The authors make use of the divergence times estimated to discuss the evolution of geographical range of fig and wasp species. In the absence of any closely related fossil, I doubt that it is possible to obtain estimates that are precise enough for this, but in any case I do not believe this is a strictly part of the paper.

However, if the authors do opt to continue relying on a calibrated phylogenetic tree, several problems with the current tree must be corrected.

The calibration point used is not well-justified. The crown age of the 4 species of *Blastophaga* included in Cruaud et al. is 15 million years, not 10 million years as used here. Therefore, one assumes that the node used for calibration is the split between *B. nipponica* and two undescribed species of *Blastophaga*, obtained from *Ficus stenophylla* and *F. chapaensis*. Since neither of these are included, it is not clear why this node in Cruaud et al corresponds to the crown group of the species sampled here. For example, if all species here are actually more closely related to *B. nipponica* than to any other species sampled in Cruaud et al, then this would be overestimating the age of the group. Alternatively, if there are species included here that do not belong to the clade in Cruaud et al, this would be underestimated. One indication that something is wrong is that the split between *Wiebesia* and *Blastophaga* was inferred to ~20 million years (Fig S5), half of the age of the equivalent node in Cruaud et al. In fact, this makes me wonder why the splits between genera were not used as calibration points, since their correspondence between analyses seems much more straightforward.

Another problem with the analysis is the topological prior used (Yule prior). This assumes no extinction, complete sampling, and that tips correspond to species. Because multiple samples per species are used, the latter is violated. Complete sampling is also certainly violated. To accommodate the multiple samples per species, I would suggest clustering sequences in species corresponding to the clades identified and using clades as tips in an analysis using the multispecies coalescent model in STARBEAST2 instead of a concatenated analysis with each sample as tip. Together with a more appropriate prior (for example, birth-death) and using the splits between genera as calibration points, this should produce more reliable estimates.

Finally, there is not need to assume a GTR+GAMMA model in STARBEAST2 (RAxML only has that option), and the substitution model can be inferred together with the tree with the BEAST2 package `bmodeltest`

(see <https://github.com/BEAST2-Dev/bModelTest/wiki>)

Reply: Thank you for your kind comments. We have re-analyzed the divergence time of

the fig wasp clades according to your suggestions.

In this reanalysis, we added more outgroup species to the analysis for the use of calibration points. As a result, three calibration points following Cruaud et al.²⁸ were used to estimate the divergence time of fig wasps; split between *Blastophaga* and *Wiebesia* (~40.5 Mya), split between *Platyscapa* and *Eupristina* + *Pegoscapus* (~43.5 Mya), and crown of Agaonidae (~75.1 Mya).

Please refer to the section, *Analysis of pollinator divergence times* in Methods (Lines 716-736) and Fig. S8.

4) Species delimitation

Especially for wasps, the authors split groups in the phylogeny arbitrarily as "clades". It seems to be this is a missed opportunity for a more reproducible analysis or criterion that clearly identifies what are these clades. In line 246, for example, the authors discuss that the % difference between samples is close that what is typically considered as different species. Are they different species of fig wasps? I would suggest the usage of tree-based species delimitation methods instead of arbitrary definition of clades. Some examples of possible methods are:

STACEY (BEAST2

package: <https://www.beast2.org/2015/06/02/species-delimitation-with-beast.html>)

Jones G. Algorithmic improvements to species delimitation and phylogeny estimation under the multispecies coalescent. *J Math Biol.* 2017 Jan;74(1-2):447-467. doi: 10.1007/s00285-016-1034-0. Epub 2016 Jun 10. PMID: 27287395.

GMYC (for example, available in <https://species.h-its.org/gmyc/>)

Tomochika Fujisawa, Timothy G. Barraclough, Delimiting Species Using Single-Locus Data and the Generalized Mixed Yule Coalescent Approach: A Revised Method and Evaluation on Simulated Data Sets, *Systematic Biology*, Volume 62, Issue 5, September 2013, Pages 707–724, <https://doi.org/10.1093/sysbio/syt033>

With this delimitation, it would be easier to discuss the patterns observed, and the species limits could also be used for a multispecies coalescent model for the calibrated tree as I suggest above.

Reply: Thank you for your kind suggestions.

We have added an analysis about species delimitation based on the phylogenetic tree of pollinator wasps using the Poisson Tree Processes (PTP) model

(<https://species.h-its.org/ptp/>). The methods (Lines 780-787) and results (Lines 222-236) are described in the text of the revised manuscript (Please refer to Fig. S7). The results are also used for discussion (Lines 344-347).

5) Biogeography and cospeciation

Just like with species delimitation, the discussion of biogeographical implications of the phylogenetic trees presented relies on arbitrary interpretation rather than reproducible methods. I suggest that the authors used their ultrametric phylogenetic trees in a program such as BioGeoBEARS (see <https://github.com/nmatzke/BioGeoBEARS>) or RevBayes (see <https://github.com/nmatzke/BioGeoBEARS>) to infer the biogeographical history of figs and wasps. Even if no reliable calibration is possible and the ultrametric tree has branch lengths in units of substitution per site, this would be better than arbitrary interpretation.

For example, I believe it would show migration of wasps not only between mainland and islands (the only ones in Fig. 6), but also between islands and the mainland, since *B. yeni* corresponds to a paraphyletic group of two species.

The same can be said about inferences on host shifts. See RevBayes (https://revbayes.github.io/tutorials/host_rep/host_rep.html) and eMPress (<https://sites.google.com/g.hmc.edu/empress/home>) for cophylogeny reconciliation.

Reply: Thank you for your kind suggestions.

We have added an analysis for cophylogeny reconciliation between the phylogenetic relationships of figs and pollinators using eMPress (<https://sites.google.com/g.hmc.edu/empress/home>). Please refer to Methods section, *Cophylogenetic analysis* (Lines 789-799). The results are described in the Results section, *Cophylogenetic analysis between figs and pollinators* (Lines 312-322, and Fig. S10), and also used for relevant discussion in the sections, *Establishment of *F. erecta* complex and their pollinators in Taiwan-Ryukyu islands* and *Host-specificity reestablishment leading to host-shift speciation*.

6) Introgression

In lines 270-274, the authors discuss whether or not sharing of fig pollinators could lead to introgression between fig species. The incongruence between trees from plastid genes and nuclear reduced-representation data may indicate that there is some degree of introgression but this is not evaluated in this paper. If MIG-seq is similar to other reduced representation libraries, it should provide enough data to address this question. For example, a recent paper also found evidence for host switches in fig wasps and in

that case they led to hybridization between fig species. This paper is not currently discussed in this manuscript, but it seems to be relevant: Wang, G., Zhang, X., Herre, E.A. et al. Genomic evidence of prevalent hybridization throughout the evolutionary history of the fig-wasp pollination mutualism. *Nat Commun* 12, 718 (2021). <https://doi.org/10.1038/s41467-021-20957-3>.

I suggest the authors to consider whether a similar kind of analysis could be performed here.

Reply: Thank you for your kind suggestions.

As you pointed out, some inconsistent results were found between plastid DNA and nuclear DNA data (SSR), and that might be due to introgression by hybridization between figs. A certain incongruence is that the TW population of *F. erecta* was grouped with GD/HK populations in plastid DNA network, while it closely related to FJ population in population genetic analysis based on SSR data. The most reasonable explanation would be that the pollinator wasps (*B. nipponica*) carrying pollen dispersed from Taiwan to the Fujian area, and hybridization of the figs between the two areas has occurred through the dispersal of pollen but not seeds. We have added discussion about this point to the revised manuscript. Please refer to **Lines 468-483 and Lines 521-533**.

On the other hand, as you may see from the network (Fig. S2) of plastid DNA that three genetically distinct clusters can be confirmed, but the sequence data could not give detailed and robust phylogenetic relationships among the fig species and populations due to very small sequence differences. Therefore, we did not add the analysis such as done in Wang, G., et al. (2021), because the analysis should be based on detailed and robust phylogenetic relationships. As well as, in our results, the certain incongruences between plastid DNA and SSR data are very simple and could be reasonably interpreted.

Wang, G., et al. (2021) is an excellent study. The evidence of prevalent hybridization between fig species imply that host-shift event might have occurred often and prevalently during the evolutionary history of the fig-wasp pollination mutualism. We have mentioned this study in our revised manuscript. **Lines 540-544**.

7) General formatting

The analysis of pollen in wasp bodies mentioned in the discussion in line 269 is not described in methods or results.

Reply: Thank you for your kind comments.

We have added the methods (**Lines 652-660**) and results (**Lines 167-183**) to the revised

manuscript.

I believe the journal instructs authors to finalize the introduction with a summary of major results.

Reply: Thank you for your kind comments.

We have made a major revision to the introduction. Please refer to the revised introduction.

Minor comments:

line 51

"specious" - probably meant "speciose"

Reply: We have corrected.

line 115 since there were so many different types of sequencing methods used, I suggest that these numbers are presented in a supplementary table and that the method that corresponds to these reads is stated explicitly (MIG-seq). More than raw reads, I would be interested in knowing the number of sites (which is reported) and the amount of missing data in the matrix (not reported currently). Is missing data biased with respect to phylogeny, as in other reduced representation methods such as RADseq?

Reply: Thank you for your kind comments.

The MIG-seq sequencing and SNP detection were outsourced to a specialized company (Bioengineering Lab. Co., Ltd., Kanagawa, Japan). The company did not provide us the information such as you suggested. So, I am sorry that we could not respond satisfactorily to the points you pointed out. However, we can provide you the SNP alignment if it is necessary.

line 181 - why not include it as supplement?

Reply: We have included it as supplementary data (Fig. S9).

line 269 - please add details in methods and results

Reply: Thank you for your kind suggestions.

We have added the descriptions about the analysis of ITS data in method section, *PCR of plastid DNA and ITS sequences of figs* (Lines 652-660) and result section, *Analysis of ITS sequences of fig pollen carried by foundresses* (Lines 167-183).

line 334 - see comments on divergence dating. Regardless, the confidence interval is large enough to encompass this event

Reply: Thank you for your kind comments.

We have re-analyzed the divergence dating according to your suggestions listed above (please also refer to the reply). The results and relevant discussion have been revised.

Lines 238-247, Lines 417-420, Lines 444-453, Lines 459-467.

line 485 - SNs - probably meant SNP?

Reply: We have corrected.

Fig 4 - This figure is really hard to interpret. Colors are confusing since the same colors do not correspond to the same populations. I would suggest to use similar colors for similar locations to the extent possible. This would make comparison of Fig 4a and 4d easier, for example.

I would also suggest a single value of K in Fig. 4b. Maybe the other values could be shown in a supplementary figure.

Reply: Thank you for your kind comments.

We have revised this figure according to your suggestions. We hope the revised version is easier to understand. Please refer to Fig. 4 and Fig. S9.

Fig. 5 - This is a nice conceptual image, but the migration and host shift events are based on arbitrary interpretation rather than reproducible analyses, see comments above.

I suggest redoing this figure after results of biogeographical and cophylogeny analyses

Reply: Thank you for your positive comments.

We have performed additional analyses according to your suggestions. Please refer to Fig. S10.

Fig. 6 - same as Fig. 5

Reply: In this figure, a-d just give the detailed explanations based on Fig. 5, and e shows the proposed model of host-shift speciation. We have made some minor changes and added a more detailed description to the caption.

Fig. S1 - Please use the same colors for species in panels A and B

Reply: Thank you for your kind suggestions.

For this figure, the phylogenetic tree does not make much sense because the supporting values for tree nodes are very low. This is due to too small sequence differences as

shown in the haplotype network. Therefore, in the revised version, we have removed the tree and left only the network to show the genetically distinct clusters and haplotypes. In addition, the haplotypes are indicated by the numbers h1-h24. Please refer to Fig. S2.

Fig. S4 - Please indicate the ingroup (and mention in the appropriate section in the methods that outgroups were used and how they were chosen)

Reply: Thank you for your kind comments.

The outgroups were indicated on each tree. The use and selection of outgroups were mentioned in the method sections, *MIG-seq experiment for figs* (Lines 617-634) and *PCR of 28S rDNA, ITS1, and COI-COII sequences in pollinators* (Lines 667-675).

Fig. S5 - Please add a time axis in bottom, which makes visualization easier

Reply: Thank you for your kind suggestions.

We have performed the dating analysis according to your suggestions listed above. And we also added more outgroup species into this analysis. A time axis was added in bottom of the figure. Please refer to Fig. S8.

Tables S5-S6 - maybe 3 decimal places would suffice and make more readable?

Reply: We have revised to be 3 decimal places. Please refer to Tables S2 and S3.

Reviewers' comments:

Reviewer #1 (Remarks to the Author):

The Authors did a lot of work according to the comments from three reviewers. I have no more comments.

Reviewer #2 (Remarks to the Author):

The revisions have satisfied my original concerns. The authors should be commended for the scale of their study, it represents an important contribution to our understanding of the evolution of the fig/fig wasp mutualism as well as other strict brood site mutualisms.

Reviewer #3 (Remarks to the Author):

I commend the authors for the careful revision of this paper, which made the story much more clear and the results much more robust. The authors did a great job in using several lines of evidence to untangle a complicated system and robustly demonstrate the details of how host shifting and pollinator sharing can differentially affect the evolution of figs and wasps. Given that much of the literature still focus on the extreme specialization and ideas on strict co-speciation, it is definitely a significant contribution.

Almost all comments that I made previously have been addressed directly. The only significant exception is the lack of a formal analysis of biogeography. However, considering the context of this version of the paper, I believe it is not longer necessary to improve clarity and reproducibility.

The only part that still gets me confused is why the authors treat "B. taiwanensis" as a different entity from "B. nipponica" if they are indistinguishable. It seems this is a case of pollinator sharing in Taiwan, similar to *B. silvestriana* in the mainland. But both cases are treated completely differently in the conceptual Figs. 5 and 6, and in the discussion. Is there any evidence that the complex *B. taiwanensis*/*B. nipponica* in Taiwan is not the same pool of wasps pollinating two species of figs?

It seems that the pollen barcoding analysis was not carried out in Taiwan, so it cannot serve as evidence in that case. Are there strong morphological differences between *B. taiwanensis* and *B. nipponica*? This could suggest, but not prove, that we are talking about two specialized but very recently diverged populations. If this evidence exists, I believe it should be cited in the discussion. But if the wasps cannot be distinguished by any trait other than the plant in which they have been sampled, I believe this calls for a different interpretation of results. If there is no evidence that *B. taiwanensis* and *B. nipponica* from Taiwan are different at all, why treat them as a case of incipient specialization and speciation? With the evidence presented currently, to me it seems more likely to be a case of a generalist wasp shared by two species of figs with some previous taxonomic confusion resulting in more than one name being in use. Because this changes the interpretation of results significantly, I consider that this clarification is still a major point to be addressed.

Other than that, I have a few minor suggestions:

lines 137-138 - the sentence suggests that *F. erecta* is not monophyletic in the tree, but Fig. 2 shows that it just does not have strong support. I suggest rephrasing to clarify. Something like: "... showed that all of the fig species formed monophyletic groups, but with low support for *Ficus erecta*"

lines 138-142 - I believe the description of phylogenetic relationships between species can be safely deleted, since Fig. 2 describes it quite well already. I would leave the sentences about splits within species, which highlight a connection to geography that is not necessarily obvious.

line 161 - Fig. 4 is mentioned before Fig. 3, so it would be appropriate to reorder them.

line 197 - I suggest only start using the double quotes after their meaning is explained in line 202. Otherwise the meaning of the quotes is not clear. Following up on my major comment above, if the two wasps are considered to be the same species, I suggest stop using "*B. taiwanensis*" entirely in the discussion, after it is established that it is just a synonym of *B. nipponica*, and adding that information to the abstract.

Data availability - do the accession numbers correspond to MIG-seq only? It is not clear what is deposited where, and I could not find DRR315461 by searching NCBI. I would also suggest adding the calibrated phylogenetic tree in nexus format to the dryad archive to enable reuse of the results.

Reply: First of all, we would like to thank the reviewers for their valuable comments and suggestions that have helped us to significantly improve our paper.

Reviewers' comments:

Reviewer #1 (Remarks to the Author):

The Authors did a lot of work according to the comments from three reviewers. I have no more comments.

Reply: Thank you for your positive review of our revised manuscript.

Reviewer #2 (Remarks to the Author):

The revisions have satisfied my original concerns. The authors should be commended for the scale of their study, it represents an important contribution to our understanding of the evolution of the fig/fig wasp mutualism as well as other strict brood site mutualisms.

Reply: Thank you for your positive review of our revised manuscript.

Reviewer #3 (Remarks to the Author):

I commend the authors for the careful revision of this paper, which made the story much more clear and the results much more robust. The authors did a great job in using several lines of evidence to untangle a complicated system and robustly demonstrate the details of how host shifting and pollinator sharing can differentially affect the evolution of figs and wasps. Given that much of the literature still focus on the extreme specialization and ideas on strict co-speciation, it is definitely a significant contribution.

Almost all comments that I made previously have been addressed directly. The only significant exception is the lack of a formal analysis of biogeography. However, considering the context of this version of the paper, I believe it is not longer necessary to improve clarity and reproducibility.

The only part that still gets me confused is why the authors treat "*B. taiwanensis*" as a different entity from "*B. nipponica*" if they are indistinguishable. It seems this is a case of pollinator sharing in Taiwan, similar to *B. silvestriana* in the mainland. But both

cases are treated completely differently in the conceptual Figs. 5 and 6, and in the discussion. Is there any evidence that the complex *B. taiwanensis*/*B. nipponica* in Taiwan is not the same pool of wasps pollinating two species of figs?

It seems that the pollen barcoding analysis was not carried out in Taiwan, so it cannot serve as evidence in that case. Are there strong morphological differences between *B. taiwanensis* and *B. nipponica*? This could suggest, but not prove, that we are talking about two specialized but very recently diverged populations. If this evidence exists, I believe it should be cited in the discussion. But if the wasps cannot be distinguished by any trait other than the plant in which they have been sampled, I believe this calls for a different interpretation of results. If there is no evidence that *B. taiwanensis* and *B. nipponica* from Taiwan are different at all, why treat them as a case of incipient specialization and speciation? With the evidence presented currently, to me it seems more likely to be a case of a generalist wasp shared by two species of figs with some previous taxonomic confusion resulting in more than one name being in use. Because this changes the interpretation of results significantly, I consider that this clarification is still a major point to be addressed.

Reply: Thank you for your kind remarks.

For “*B. taiwanensis*”, there are two important points that are different from the case of *B. silvestiana*. 1) the “*B. taiwanensis*” was established by host shift of *B. nipponica* from *F. erecta* to *F. formosana* that migrated from mainland China to Taiwan. These processes, namely the migration of host fig and the host shifting of pollinator, are the same as the initial stages of the speciation of *B. nipponica*. 2) Population genetic analysis showed that there was significant population genetic differentiation between the pollinators *B. nipponica* and “*B. taiwanensis*” (Supplementary Table S3) although they are indistinguishable in the phylogenetic tree. In addition, despite of extensive sampling, we did not find any evidence of hybridization between their host figs by morphological observations and population genetic analysis of SSR data (Fig. 4b). These findings suggest that the host-specificity between *F. formosana* and “*B. taiwanensis*” may have proceeded to some extent, leading to pre-mating isolation between *B. nipponica* and “*B. taiwanensis*” as well as between their host figs (*F. erecta* and *F. formosana*) by restricting pollen flow. Therefore, we think that it would be appropriate to discuss the “*B. taiwanensis*” in the same section as *B. nipponica*.

As mentioned above, the host-specificity in fig-pollinator obligate mutualism will bring pre-mating isolation between pollinators as well as between their host figs. Since the

present results have implied that the host-specificity between *F. formosana* and “*B. taiwanensis*” may have proceeded to some extent, we think that it is better to do the process for whether to treat “*B. taiwanensis*” as a synonym of *B. nipponica* after conducting host-discrimination experiments and morphological reexamination of the “*B. taiwanensis*”. We have made a plan for this future research. Once the COVID-19 infection subsides, we plan to carry out the study.

According to your suggestions, we have made some revisions to the discussion on “*B. taiwanensis*”. **Lines 517-543.**

We have also changed [“Speciation” by host-shift] to [Host-shift] in Fig. 6c.

Other than that, I have a few minor suggestions:

lines 137-138 - the sentence suggests that *F. erecta* is not monophyletic in the tree, but Fig. 2 shows that it just does not have strong support. I suggest rephrasing to clarify. Something like: "... showed that all of the fig species formed monophyletic groups, but with low support for *Ficus erecta*"

Reply: Thank you for your kind suggestion. We have revised. **Line 137.**

lines 138-142 - I believe the description of phylogenetic relationships between species can be safely deleted, since Fig. 2 describes it quite well already. I would leave the sentences about splits within species, which highlight a connection the connection to geography that is not necessarily obvious.

Reply: We have deleted the description. **Line 137.**

line 161 - Fig. 4 is mentioned before Fig. 3, so it would be appropriate to reorder them.

Reply: We have changed (see Fig. 4a and 4b) to (see the results in *Population genetic differentiation of figs* below), and kept the order of figures. **Lines 157-158.**

line 197 - I suggest only start using the double quotes after their meaning is explained in line 202. Otherwise the meaning of the quotes is not clear. Following up on my major comment above, if the two wasps are considered to be the same species, I suggest stop using "*B. taiwanensis*" entirely in the discussion, after it is established that it is just a synonym of *B. nipponica*, and adding that information to the abstract.

Reply: We have revised the description. **Lines 193-202.** Please also refer to our response to your comments above.

Data availability - do the accession numbers correspond to MIG-seq only? It is not clear what is deposited where, and I could not find DRR315461 by searching NCBI. I would also suggest adding the calibrated phylogenetic tree in nexus format to the dryad archive to enable reuse of the results.

Reply: We have added the accession numbers for all sequence data. Hold-Date for MIG-seq was set to 2022-02-24, and that for other sequence data was set to 2022-02-20 by default. If necessary, we could ask DDBJ to change the Hold-Date to public immediately. Also, we have added the calibrated phylogenetic tree in nexus format to the dryad archive. **Lines 812-818.**